# Strato-structural evolution of the deep-water Orange Basin: Constraints from 3D seismic reflection data

Nombuso G. Maduna[1], Musa S.D. Manzi[1], Zubair Jinnah[1], Julie E. Bourdeau[2]

[1]School of Geosciences, University of the Witwatersrand, Johannesburg, PBag 3, WITS, 2050, Republic of South Africa
5  [2]Geological Survey of Canada, 601 Booth Street, Ottawa, Ontario, K1A 0E8, Canada.

*Correspondence to*: Nombuso G. Maduna (nombuso.maduna@gmail.com)

**Abstract.** Deep-water fold-and-thrust belt (DWFTB) systems are gravity driven collapse structures often found in passive margin settings and are comprised of a linked up-dip extensional domain, central transitional/ translational domain, and down-dip compressional domain. Many Late Cretaceous DWFTB systems occur along the SW African passive margin with multiple, 10  over-pressurized, seaward-dipping shale detachment surfaces accommodating gravitational slip. In this study we use 3D reflection seismic data to constrain the strato-structural evolution of the translational and compressional domains of a Late Cretaceous DWFTB system and the overlying Cenozoic deposits in the Orange Basin, South Africa. The stratigraphy and structure of the Late Cretaceous DWFTB system is shown to have controlled fundamental sedimentary processes and the stability of the evolving margin. The compressional domain exhibits large-scale landward-dipping DWFTBs with thrust faults 15  detaching the main Turonian shale detachment surface at depth and terminating at the early Campanian surface. A major ~7 km wide seafloor slump scar reflecting margin instability occurs directly above a syncline of the same width from the buried DWFTB system's compressional domain. The translational domain is imaged as a complex region displaying overprinted features of both extensional and compressional tectonics with the downslope translation of sediment comprising listric normal, then thrust and oblique-slip faults distally. Thrust sheets are segmented along strike by extensive oblique-slip faults which 20  extend from the translational domain into the down-dip compressional domain. Smaller, localized fold-and-thrust belts are found directly below the kilometre-scale DWFTB system in the down-dip compressional domain detaching a lower, Albian shale detachment surface which corresponds to an older gravitational collapse. The upward propagation of normal and oblique-slip faults with progressive sedimentation is hindered by the Oligocene or Miocene stratigraphic markers corresponding to mass erosional processes in the Cenozoic. A large (~2.3 km wide), roughly slope-perpendicular Oligocene submarine canyon 25  formed by turbidity currents is attributed to a major sea-level fall at ~30 Ma. Oceanographic circulation is shown to have held a significant control on the deposition of mid-Miocene to present-day sedimentary sequences. Between 1 200 to 1 500 m water depths along the upper continental slope well-preserved extensive slope-parallel, sinusoidal channel-like features occur on the Miocene stratigraphic marker. The channels are confined within a ~14 km wide zone at the interface of the upper northward flowing Antarctic intermediate water (AAIW) and deeper southward flowing North Atlantic Deep Water (NADW) currents. 30  The erosive interaction of these opposite flowing bottom currents combined with the effects of the Benguela Upwelling System (BUS), all of which formed or intensified at ~11 Ma, are responsible for the creation and preservation of the extensive slope-parallel channels.  This study shows the difference in structural styles of the translational and compressional domains of a Late Cretaceous DWFTB system and the processes responsible for mass-scale erosion in the Cenozoic.

## 1 Introduction

The southwest African margin is a prime site for the study of processes in passive margin settings. From shallow to deep-water environments, the margin not only records the Late Jurassic to Early Cretaceous break-up of Gondwana, but also its post-rift evolution (Fig. 1; Séranne and Anka, 2005). The gravitational collapse and subsequent contraction of sedimentary sequences in the deep ocean formed deep-water fold-and-thrust belts (DWFTBs); distal sedimentary wedges of interrelated folds and

thrust faults over a sloping detachment surface (Fig. 2; Rowan et al., 2004; Nemčok et al., 2005). DWFTBs form part of a linked tripartite system consisting of a: 1) down-dip compressional domain (within which DWFTBs form), 2) central transitional/ translational domain, and 3) up-dip extensional domain with proximity to the coastline (Fig. 3; Rowan et al., 2004; Corredor et al., 2005; de Vera et al., 2010; Morley et al., 2011). The advancement in deep-water drilling technology by the petroleum industry over the last two decades has promoted interest in the study of DWFTBs, as their anticlines are known to host vast reserves of hydrocarbons (e.g., Bilotti and Shaw, 2005; Corredor et al., 2005).

Well-preserved DWFTB systems are found within the Orange Basin which is located offshore South Africa and Namibia (Figs. 2a and 3; Butler and Paton, 2010; de Vera et al., 2010; Scarselli et al., 2016). The use of 2D reflection seismic data acquired in the Orange Basin has allowed certain key elements of its evolution to be constrained through several studies (e.g., Light et al., 1993, Clemson et al., 1997; de Vera et al., 2010; Dalton et al., 2015, 2017; Collier et al., 2017; Baby et al., 2018). These studies describe events within the late stages of continental breakup, to the stratigraphy, structure, and hydrocarbon potential of the basin, with later studies including the formation of gravitational collapse structures (DWFTB systems). The observations are however limited for the South African region of the Orange Basin, particularly in the distal deepwater environment which has been poorly explored in contrast to the Namibian extent. Directly adjacent to the South African maritime border (and hence this present study), significant light oil discoveries from deepwater wells drilled between 2 000–3 000 meters below sea level (mbsl) have been reported in Namibia (Fig. 2; van der Spuy and Sayidini, 2022). Although there is extensive 2D legacy seismic data coverage, the deepwater Orange Basin in South Africa remains largely underexplored due to the sparsity of wells (Fig. 2), and both 2D and 3D seismic data coverage (van der Spuy and Sayidini, 2022). According to the Petroleum Agency of South Africa (PASA hereon), only 38 exploration wells have been drilled with one well per 4 000 km$^2$, most of which are confined to the shelf environments below 1 000 mbsl (Fig. 2; PASA, 2017; van der Spuy and Sayidini, 2022). Dense 2D legacy seismic data coverage is confined to the shelf while the more distal, deepwater environment has fewer and farther-spaced seismic lines. Furthermore, only three 3D seismic surveys have been conducted in the deepwater environment (van der Spuy and Sayidini, 2022).

There is a great paucity of data for the central transitional/translational domain of gravity collapse structures both offshore SW Africa and worldwide due to its structural complexity and how it has previously been poorly seismically imaged (mainly 2D), thus making it difficult to interpret (e.g., Butler and Paton, 2010). In stark contrast, there is extensive knowledge on the up-dip extensional and down-dip compressional domains, which are well-studied due to the simplicity of the former and known hydrocarbon potential of the latter (e.g., Butler and Paton, 2010; de Vera et al., 2010; Scarselli et al., 2016). An examination of the translational domain in a buried DWFTB system is done in this study using one of the existing deepwater 3D reflection seismic surveys (Fig. 2). We define its relationship to the down-dip compressional domain and assess how both domains have structurally affected the younger deposits following progressive sedimentation. We also look at the along-strike change in stratigraphy and structure of both domains along the margin. From these observations we create a model describing not only

the evolution of one DWFTB system, but also that of the overlying sediments and their erosional features as affected by global oceanographic circulation.

## 2 Regional setting

### 2.1 Offshore structural framework

As in all margins, very little is known of the SW African margin's configuration prior to rifting (Mohammed et al., 2017). What is known is that the pre-rift basement forms a 30 km wide, N-S oriented zone of pronounced flexure induced by thermal subsidence in the continental lithosphere (Light et al., 1993; Clemson et al., 1997; Mohammed et al., 2017; Baby et al., 2018). This zone, known as the hinge line, has separated regions of subsidence with regions of stasis or uplift ever since the Mesozoic, forming a critical boundary of the margin's offshore and onshore morphologies (Light et al., 1993; Clemson et al., 1997;
Aizawa et al., 2000). The hinge line forms a N–S structural grain offset by several E–W oriented segment boundaries having partitioned the timing of rifting in continental break-up (Clemson et al., 1997). The E–W oriented segment boundaries are fracture zones which accommodated the zipper-like, south to north development of Early Cretaceous depocentres in zones of greatest subsidence (Light et al., 1993; Baby et al., 2018). From oldest to youngest these depocentres mark the position of the Outeniqua, Cape, Orange, Lüderitz and Walvis basins contained between the Rio Grande Fracture Zone to the north, and the
Aghulhas-Falkland Fracture Zone to the south, which enabled right-lateral (dextral) strike-slip motion during rifting (Fig. 1; Séranne and Anka, 2005). By analogy of the southern Outeniqua Basin, rifting and continental break-up likely began around the Middle Jurassic (~160 Ma) up until ~131 Ma in the Early Cretaceous (Dingle et al., 1983; Collier et al., 2017). This first stage of margin deformation (extension and rifting) was followed by continental drift with the onset of a spreading oceanic ridge subdividing the margin into well-defined shelf, slope and basinal environmental settings (Fig. 1; Light et al., 1993;
Séranne and Anka, 2005).

### 2.2 Offshore stratigraphy

The syn-rift succession comprises of sediments deposited during extension and rifting of the margin, while the post-rift succession comprises of sediments deposited since continental drift until present (Fig. 4; e.g., de Vera et al., 2010). Sedimentation and facies distribution were tectonically controlled in the deposition of the syn-rift succession (Light et al.,
1993), which forms a megasequence of Late Jurassic to Early Cretaceous (late Hauterivian) siliciclastic and volcaniclastic sediments (Fig. 4, Paton et al., 2008; Dalton et al., 2017). The syn-rift succession is a classic example of a volcanic rifted margin setting as it is characterised by seaward dipping reflectors (SDRs) reflecting a large subaerial, basaltic extrusive event associated with during the break-up of the margin (Maslanyj et al., 1992; Menzies et al., 2002; Baby et al., 2018). The SDRs form a thick (>3 km) package (Baby et al., 2018) comparable in age to the Parana-Etendeka Large Igneous Province at ~135
Ma (Koopman et al., 2016; Collier et al., 2017). Sedimentation and facies distribution were tectonically controlled during the

rifting phase, hence the presence of these SDRs together with rotated and eroded extensional fault blocks in the syn-rift succession (Maslanyj et al., 1992; Light et al., 1993; Granado et al., 2009). The oldest evidence of mid-oceanic ridge (MOR) activity and the transition to oceanic crust occurs at the M3 magnetic anomaly (Figure 2.2), between the Hauterivian and Barremian sequences at 127 Ma (Séranne and Anka, 2005; Collier et al., 2017). Overall, the syn-rift succession is comprised

of isolated half grabens infilled with interbedded clastics and volcanics (Jungslager, 1999). The change from subaerial conditions during rifting to an open marine environment with continental drift occurs at ~131 Ma with the initial deposition of the post-rift succession above the regional, late Hauterivian break-up unconformity (Figs. 3 and 4; Menzies et al., 2002; Granado et al., 2009; De Vera et al., 2010).

The Orange Basin covers an extensive area of approximately 160 000 km$^2$, including the Namibian extension, making it South
Africa's largest offshore basin both aerially and volumetrically (Kuhlmann et al., 2010; PASA, 2017). The basin has acted as the depocentre for sediments deposited from the Olifants and Orange rivers and their ancestral equivalents since the Early Cretaceous (Fig. 1; Maslanyj et al., 1992; Paton et al., 2008; de Vera et al., 2010). The ~131 Ma to present day post-rift clastic sediments currently reach a central thickness of 3 km in the south and 5.6 km in the north of the Orange Basin (Maslanyj et al., 1992; Paton et al., 2008; Granado et al., 2009; Dalton et al., 2017). Dalton et al. (2017) subdivided the post-rift succession
into three megasequences; 1) early drift (Early Cretaceous), 2) late drift (Late Cretaceous), and 3) Cenozoic. The early drift megasequence is comprised of black shales and claystones. The late drift (within which most DWFTB systems are found) and Cenozoic megasequences are comprised of interbedded heterolithic shales and claystones. The post-rift stratigraphy of the Orange Basin has been well-described by many authors (e.g., Emery et al., 1975; Brown et al., 1995; Paton et al., 2008; Granado et al., 2009; de Vera et al., 2010; Kuhlmann et al., 2010; Dalton et al., 2017; Baby et al., 2018) who subdivided the
succession into several stratigraphic units (sequences) separated by key stratigraphic markers or bounding surfaces (Fig. 4).

### 2.3 Gravitational collapse structures

Gravitational processes in the Orange Basin have been enhanced by several phases of uplift and denudation, responsible for crustal thinning and the inversion of extensional faults recorded in the SW African margin's Cretaceous post-rift stratigraphic succession (Granado et al., 2009; de Vera et al., 2010; Hirsch et al., 2010; Hartwig et al., 2012; Wildman et al., 2015; Baby et
al., 2020). This resulted in the impressive formation of DWFTB systems observed throughout the SW African margin (Fig. 1). DWFTB systems are mass transport deposits consisting of a linked up-dip extensional domain, central transional/translational, and down-dip compressional domain, which are often well-preserved in passive margin settings (Fig. 3; Rowan et al., 2004; Bilotti and Shaw, 2005; Krueger and Gilbert, 2009). The extensional domain is characterized by convex-upward growth strata separated by listric normal faulting up-dip, adjoining onto the transitional domain down-dip (Fig. 3; Butler and Paton, 2010).
With the continual supply of sediment up-dip in passive margins, normal faulting within the extensional domain eventually leads to down-dip compression along the basal detachment, and the subsequent formation of fold and thrust belts in the compressional domain (Fig. 3; King and Morley, 2017). The anticlines of distal folds in the compressional domain are known

to host vast accumulations of hydrocarbons as proven previously in the Niger Delta (Bilotti and Shaw, 2005; Corredor et al., 2005) and recently in the northern Orange Basin of Namibia (van der Spuy and Sayidini, 2022). To form gravity collapse structures in passive margin settings, a very deep sedimentary basin with high rates of sedimentation upon a dipping slope is required (Krueger and Gilbert, 2009). Once enough overburden material has accumulated in the basin, combined with a lack of cohesion, gravitational sliding occurs along a weak, sloping detachment layer (Rowan et al., 2004; Morley et al., 2011). This gravity sliding process, together with the lateral flattening of sediment under its own weight in gravity spreading, results in gravitational collapse and the subsequent formation of DWFTBs (Morley et al., 2011). The combination of episodic margin uplift and thermal subsidence along the SW African margin resulted in downslope gravitational sliding and spreading forming DWFTB systems in the Orange Basin (de Vera et al., 2010). Trigger mechanisms for gravitational collapse and slope instability offshore SW Africa include earthquake activity (seismicity) and rapid rates of sedimentation causing elevated internal pore fluid pressures (Séranne and Anka, 2005; Kuhlmann et al., 2010).

Various classification schemes have been proposed for DWFTB systems based on the type of stress field, tectonic setting and basal detachment lithology for which the system is under (e.g., Rowan et al., 2004; Hamilton and de Vera, 2009; Krueger and Gilbert, 2009). Combining these, Morley et al. (2011) classified DWFTB's into: Type 1) near-field stress-driven systems occurring predominantly in passive margin settings; and Type 2) far-field together with mixed near- and far-field stress-driven systems occurring predominantly in active margin settings. Type 1a systems are comprised of seaward- or landward- dipping basal shale detachments, while Type 1b are comprised of seaward-dipping basal salt detachments. The Orange Basin DWFTB systems are classified as Type 1a, near-field stress-driven systems containing multiple shale detachment surfaces (Morley et al., 2011). The multiple detachments surfaces upon which gravitational sliding occurs include the Aptian, Turonian and Campanian seaward-dipping shales (Séranne and Anka, 2005; Dalton et al., 2015). The most significant, regional shale detachment surface is Turonian-aged while others are more locally developed (de Vera et al., 2020). The shales correspond to maximum flooding surfaces, marking the end of a marine transgression (Baby et al., 2018), and are associated with high fluid overpressures and source rock intervals (van der Spuy et al., 2003; de Vera et al., 2010; Kuhlmann et al., 2010). The widespread occurrence of subsurface and surface gas/fluid escape features serves is indirect evidence of over-pressurized conditions (Morley et al., 2011). Fluid overpressures are mainly formed by the combined effects of volumetric expansion involved in hydrocarbon generation and maturation, tectonic stresses and disequilibrium compaction (Rowan et al., 2004; Bilotti and Shaw, 2005).

## 2.4 Palaeoceanography

According to Uenzelmann-Neben et al. (2017), oceanic circulation since the Albian to Paleocene/early Eocene exhibited a proto-Antarctic Circumpolar Current (ACC), not strong enough to affect sedimentation in the western South Atlantic. Since the Eocene/Oligocene boundary however, a major change in oceanic circulation patterns occurred with the onset the Antarctictic Circumpolar Current (ACC) and Atlantic meridional overturning circulation (AMOC) after the opening of the

Drake Passage in the early Oligocene (~31–28 Ma). With the intensification of the AMOC and strong ACC, southern sourced bottom and deepwater currents formed in the mid-Miocene and were responsible for the change in sedimentation offshore SW Africa. The bottom and deepwater currents offshore western South Africa include the Antarctic Intermediate Water (AAIW), the North Atlantic Deepwater (NADW) and the deep Antarctic Bottomwater (AABW). The interplay of these currents, together with the Benguela Current offshore South Africa has affected sedimentation along the SW African margin since the mid Cenozoic (Weigelt and Uenzelmann-Neben, 2004, 2007a; Uenzelmann-Neben et al., 2017).

The Benguela Current drives surface water circulation offshore SW Africa in a northerly direction as the eastern portion of the South Atlantic subtropical gyre (Peterson and Stramma, 1991). Changes in the growth of Antarctic ice sheets, reflected in oxygen isotope data, have influenced long- and short-term variations within the current and its associated upwelling system (Diester-Haass et al., 1992). The extension of Benguela Current starts from Cape Point (34°S) in the south and ends at the Angola-Benguela front in Cape Frio (18°S) to the north. At 28°S, the current separates into an oceanic and a coastal branch referred to as the Benguela Oceanic Current (BOC) and Benguela Coastal Current (BCC), respectively. Through geostrophic flow, the BOC transports warm waters from the Agulhas Current (in the south) northwards and westwards, whereas the BCC transports colder waters from the proximal wind-dominated coastal region northwards, eventually encountering the warm Angola Current in the north (Stramma and Peterson, 1989).

During the middle to late Miocene, paleo-oceanographic studies record a sharp drop in oceanic $CaCO_3$ concentrations, termed as the 'carbonate crash', in all equatorial regions of major oceans including the Atlantic offshore SW Africa (cf. Diester-Haass et al., 2004). The carbonate crash offshore SW Africa was caused by an increase in clastic input from the Orange River during sea-level regressions resulting in terrigenous dilution. This was followed by an increase in biogenic activity during the late Miocene to early Pliocene, referred to as the 'biogenic bloom' (Hermoyian and Owen, 2001; Diester-Haass et al., 2004). In concordance with the global cooling trends recognised in the middle Miocene (13 Ma), surface ocean water temperatures are shown to have dropped significantly offshore SW Africa as Antarctic ice sheets expanded, thus intensifying south-easterly trade winds triggering the inflow of cold waters (Zachos et al., 2001; Rommerskirchen et al., 2011). An increase in total organic carbon content and benthic foraminifera accumulation is recorded from core samples taken from the Cape Basin's ODP sites 1085, 1086 and 1087, forming a depth transect across the upper continental slope of the SW African margin (Rommerskirchen et al., 2011). These high paleo-productivity rates are attributed to upwelling of the Benguela Current in the southern Atlantic intensifying at ~11 Ma (Diester-Haass et al., 2004; Rommerskirchen et al., 2011).

**3 Data and methods**

Three 3D seismic surveys have been conducted in the deepwater South African Orange Basin; one acquired in 2002, imaging low resolution data, and two large, higher resolution surveys acquired between 2012 and 2014 (van der Spuy and Sayidini, 2022). This study uses the northernmost 3D seismic dataset bordering the Namibian maritime licensing region (Fig. 2).

Following seismic acquisition by the Dolphin Geophysical Polar Duchess ship (Table 1), seismic processing was carried out by the Netherlands Global Processing Team (Table 2) on behalf of Shell Global Solutions International. In this study we interpreted the processed seismic data obtained from Shell.

## 3.1 Seismic acquisition and processing

Shell Global Solutions International commissioned a 3D reflection seismic survey between 2012 and 2013 in the deep-water Orange Basin (Kramer and Heck, 2014). The survey was designed in a ~NNW to SSE orientation, covering a total area of ~8 200 km$^2$ (Fig. 2b). Dual airgun arrays were used with a source volume of 0.0672 m$^3$ towed at 8 m depth with a 25 m shot point interval, and a 7 950 m long streamer with a 12.5 m group interval and 12.5 m group length. The data were recorded at a 2 ms sample rate for a total record length of 7 168 ms. A low-cut frequency of 4.4 Hz at a 12 db/Oct slope and a high-cut frequency

of 214 Hz at a 341 db/Oct slope were used during data acquisition, giving a dominant frequency of 50 Hz after the application of an anti-aliasing filter. The full survey acquisition parameters are summarised in Table 1. Pre-processing from field tape seismic data was first carried out onboard by the Dolphin Geophysical Polar Duchess ship. This involved data conversion from SEG-D to SEG-Y output and thereafter, the Netherlands Global Processing Team carried out the rest of the processing workflow, as summarised in Table 2. Processing was carried out at 4 ms from SEG-Y field tape datum through surface-related

multiple elimination (SRME) using 3D SRME and anisotropic Kirchhoff pre-stack depth migration (PreSDM) (Table 2).

## 3.2 Seismic interpretation

In this study we geologically interpret a ~1 800 km$^2$ seismic portion of interest (Fig. 2b) using the Petrel Schlumberger software. The seismic volume lies along the continental slope offshore western South Africa, between water depths of 1 000 to 2 000 mbsl (Fig. 2). The seismic interpretation workflow included: 1) loading the SEG-Y seismic data involving importing,

realizing and cropping; 2) applying volumetric-based attributes of structural smoothing and variance; 3) horizon and fault mapping through picking horizons and faults of interest and generating surfaces from them; 4) applying horizon-based attributes of influential data and edge detection; 5) velocity modelling and depth conversion; and 6) the creation of a 3D geological model with modelled faults and surfaces in the structural framework. Since no well log data was available for the correlation of stratigraphic sequences at the time of this study, seismic interpretation was carried out in the time domain.

### 3.2.1 Seismic resolution limit

With an average velocity of 2 400 ms$^{-1}$ reported for the Orange Basin (cf. Kuhlmann et al., 2010), and dominant frequency of ~20 Hz for the study, the vertical seismic resolution limit given for the ½ and ¼ dominant wavelength criteria (Yilmaz, 2001) is 60 m and 30 m, respectively. Since the data is migrated it collapses the horizontal resolution to the dominant wavelength giving a horizontal resolution limit of 120 m. Geological features smaller than the vertical and horizontal resolution limits are

indistinguishable. It is only through the application of seismic attributes that features lower than the resolution limit may be

detected. An example of this is presented in Manzi et al. (2013) where they detected additional fine-scale structures in a Witwatersrand gold mine than previously known.

### 3.2.3 Structure delineation using seismic attributes

Seismic attributes are mathematical measurements derived from the information provided by seismic data. They are often used to enhance important geological and physical properties of the seismic data by delineating faults, resolving thin beds, and identifying bright spots that could indicate hydrocarbon reservoirs (Chopra and Marfurt, 2005, 2007; Brown, 2011). Seismic attributes are dependent on the signal-to-noise (S/N) ratio of the data and therefore need to be conditioned using various filters. Schlumberger's Petrel software offers a wide variety of seismic attributes which may be applied on the whole volume or on interpreted horizons. Volumetric attributes may be applied from the onset of seismic interpretation to the entire volume while horizon-based attributes are only appliable once sufficient interpretation has taken place on a surface of interest (Brown, 2011).

Structural smoothing was first applied to the full seismic volume to condition the data. It is used to increase the S/N ratio by smoothing the seismic data's input signal through local averaging with a Gaussian filter (Randen et al., 2000). Using the structurally smoothed volume as input, the variance attribute was applied to enhance fault discontinuities throughout the seismic section. Variance is an edge enhancing attribute that measures local deviations from the seismic signal in the form of a coherency analysis (Silva et al., 2005), and is used in many studies to highlight faulted areas in place of the similar chaos attribute (e.g., Maselli et al., 2019).

To further enhance fault continuities and visualise 3D geometric variations, horizon-based attributes of edge detection and influential data were applied to the surfaces of interest. Horizon-based attributes were used in conjunction with the surface smoothing structural operation which filters out anomalous peaks from picking. Edge detection extracts an edge model to enhance discontinuities by combining the dip and dip azimuth properties and normalizing these to the local noise of the surface (Randen et al., 2000). Influential data generates a property on the data object that highlights areas of rapid 3D geometric variation which is key to ensuring sensible geometric form.

### 3.2.4 Seismic interpretation strategy

This study's seismic volume images the compressional and translational domains of a Late Cretaceous DWFTB system together with the overlying Cenozoic successions. The variance time slice shown in Fig. 5a cuts through both domains of the DWFTB system at a level where most structures may be seen in plan view at 3 424 ms TWT. Combining this with the plan view of all faults dip orientations in Fig. 5b was integral in aiding 3D seismic interpretation. Fig. 5c combines all observations in plan section to give a generalized overview of the structural framework and shows the position of regional sections Figs. 6, 7 and 8. These regional sections are used to describe the seismic stratigraphy and structural framework observed in and above the Late Cretaceous DWFTB system. Figure 6 is a full crossline section of the study area, showing a portion of the translational

domain and, more importantly, the compressional domain (Fig. 5). The regional section lies roughly perpendicular to the DWFTBs giving a clear view of their internal geometry and surroundings. Figs. 7 and 8 are inline sections showing the along-strike component of the compressional and translational domains, respectively, and are orientated perpendicular to NE-SW trending faults (Fig. 5).

The seismic volume was interpreted using the classical approach implemented by Mitchum et al. (1977) who first introduced the concept of seismic stratigraphy whereby the sequence stratigraphic framework is characterized by stratal termination patterns (i.e., downlap, onlap, toplap erosional truncation and concordance) between each seismic facies or sequence. Stratigraphic markers are bounding surfaces that separate each sequence and are created through the interplay of base sea level fluctuations and sedimentation marking the change in depositional regimes (Catuneanu, 2006). In deepwater marine settings

surfaces separating each seismic sequence, based on the difference in internal configuration patterns, include the maximum flooding surface (MFS), correlative conformity (CC) and maximum regressive surface (MRS) (Catuneanu 2006; Catuneanu et al., 2009). Continuous, dominant high amplitude reflections were picked as key stratigraphic markers in this study and named using the classification and terminology of Catuneanu (2006), as summarised in Table 3. Unconformity surfaces developed through erosion or prolonged periods of non-deposition (Catuneanu, 2002) are also identified in this study, often combining

with MFSs, CCs, and MRSs. Nine key stratigraphic markers were identified in the seismic section, coinciding with those recognized throughout the Orange Basin in previous studies (Fig. 4; Brown et al., 1995; Paton et al., 2008; de Vera et al., 2010; Kuhlmann et al., 2010; Hartwig et al., 2012; Dalton et al., 2017; Baby et al., 2018). The geological ages used for stratigraphic markers in this study were postulated from the comparison of the aforementioned past studies and published well data. Figure 3 shows that although the nomenclature used for stratigraphic markers (and stratigraphic sequences) differs between various

studies, the actual ages assigned to each are generally consistent.

To derive the approximate thickness of seismic sequences, a generalized depth conversion was carried out using average interval velocities from well logs acquired in the shallower reaches of the Orange Basin used in Kuhlmann et al. (2010). Velocities of 1 800 ms$^{-1}$, 2 000 ms$^{-1}$, 2 000 ms$^{-1}$ and 4 500 ms$^{-1}$ were assigned to the seafloor, Oligocene, Maastrichtian and Albian surfaces (named in this study), respectively, to create an interval velocity model for depth conversion in Petrel.

**4 Results**

**4.1 Seismic stratigraphy**

In the Cretaceous we identified and named the Albian, Turonian, Santonian, early Campanian, late Campanian, and Maastrichtian unconformity surfaces. These surfaces correspond to the 14At1, 15At1, possibly 16Dt, 17At1, 18 At1 and 22At1 MFSs and unconformity surfaces respectively, according to the offshore stratigraphic nomenclature developed by PetroSA

(previously Soeker) (Fig. 4; Brown et al., 1995; PASA, 2017). In the Cenozoic succession we identified the Miocene and Oligocene unconformity surfaces which were also recognised by Baby et al. (2018). The surfaces are markers that divide the

entire seismic succession into nine stratigraphic sequences, explained from the regional sections shown in Figs. 6, 7 and 8. The sequences were further grouped into four main megasequences (A-D) reflecting three major phases of margin evolution as described by Dalton et al. (2017): early drift (A), late drift (B1-C3), and Cenozoic (D1-D3) (Fig. 4). Older stratigraphic markers and sequences (below the Albian) were left uninterpreted in the regional sections shown in Figs. 6, 7 and 8, as their full sedimentary package lay mostly below the vertical limit of seismic data.

### 4.1.1 Early to late drift megasequences (A and B1-C3)

Sequence A in this study is a 400–700 m thick succession that forms a portion of the early drift megasequence. It is characterized by low to medium amplitude, chaotic (Fig. 8), subparallel to mounded, internal reflections (Figs. 6 and 7). The upper stratigraphic marker of the sequence is the medium amplitude Albian surface. Sequences B1 to C3 are grouped as the late drift megasequence. Sequence B1 downlaps the Albian surface as a 0–500 m thick unit characterized by deformed, medium to high amplitude, chaotic and mounded internal reflectors (Figs. 6, 7 and 8). The upper bounding surface of sequence B1 is the medium to high amplitude undulatory and irregular Turonian surface. The Turonian surface often merges with the deeper Albian surface in the translational domain (Fig. 8).

Sequence B2 forms a sedimentary wedge that thickens from 900–1 700 m seawards (down-dip direction), with the greatest thickness in the central region of the entire seismic volume, as shown in Figs. 6 and 7. Sequence B2 is characterized by deformed, medium to high amplitude seismic reflections downlapping the Turonian surface. The internal geometry of the unit in the compressional domain appears as stacked, steeply NE-dipping reflections (Fig. 6) which flatten towards the translational domain to become parallel with the underlying and overlying sequences (Fig. 8). The steeply dipping reflections are folded into asymmetric anticlines distinctly separated by imbricate thrusts creating fold-and-thrust belts (Section 4.2; Figs. 6 and 9). The upper bounding surface of sequence B2 is the Santonian surface that defines the main crest of the kilometre-scale DWFTBs. It is a thick, high amplitude surface conformable to the folded geometry of the unit. Figure 9 shows the Santonian surface in 3D and the underlying sequences. The horizon-based attribute of dip angle was computed for the surface to enhance the morphology of the fold crests dipping between 0–15°. In the NW portion of Fig. 7, towards the translational domain (see Fig. 5), a large 6.3 km wide syncline is imaged in sequence B2. To the SE the rest of the sequence appears as a shallow and very broad anticline.

Onlapping the Santonian surface is sequence C1; a sedimentary wedge that thins from 1 000 m in the translational domain to 120 m in the distal contractional domain (Fig. 6). It is characterized by low amplitude, concave-upwards sag geometries that onlap and downlap the Santonian surface (Figs. 6, 7 and 8). In the more distal region of compressional domain however, the reflectors have stacked conformably with the underlying sequence (SW portion of Fig. 6). The early Campanian upper bounding surface of sequence C1 erosionally truncates the unit (Figs. 6 and 7). Sequence C2 is a ~200 m thick unit of thin, wavy to parallel, high frequency, high amplitude seismic reflectors (Figs. 6, 7 and 8). Reflectors are initially wavy at the base

of the sequence, due to folding above the underlying thrust planes, then gradually straighten to parallel continuous reflections at the top of sequence C2 (Figs. 6 and 7). The sequence's late Campanian upper bounding surface is a thick, very high in amplitude surface, and appears conformable. Above this lies sequence C3, a seaward-thinning wedge of sediment decreasing from 200–90 m (Figs. 6, 7 and 8). The internal geometry of sequence C3 consists of thick, parallel, very high amplitude seismic reflectors, including that of its upper Maastrichtian bounding surface which appears conformable.

### 4.1.2 Cenozoic (D1-D3)

Influential data and edge detection horizon-based attributes were used to enhance the 3D geometric variation of erosional features observed upon the Oligocene and Miocene Cenozoic stratigraphic markers (Fig. 10). The lowermost Cenozoic sequence corresponds to the D1 forms a 450–100 m seaward-thinning wedge (Fig. 6). It is characterized by low amplitude, parallel seismic reflections which downlap the Maastrichtian surface mostly in the translational domain (Fig. 8). Sequence D1 (and a small upper portion of the underlying C3 in the translational domain, Fig. 8) is erosionally truncated by a SE–NW trending submarine canyon running perpendicular to the slope (Fig. 10a, b). From the observation of the transitional domain regional section (Fig. 8), the canyon is situated above a ~1.3 km wide horst from the Late Cretaceous B2 sequence with opposite dipping normal faults. The horst forms a large antiformal anticline with chaotic internal reflectors observable in sequence B2 and outlined in Fig. 8. The canyon is ~2.3 km wide with a visible length of 13 km, which extends beyond the seismic dataset (Fig. 10a, b). Onlapping against the walls of the canyon and overlying the Oligocene surface is the 100–200 m thick D2 sequence (Fig. 8). It comprises low amplitude, parallel reflectors with some chaotic sections. More than one erosional event is evident both before and after the main Oligocene unconformity surface as reflectors within sequences D1 and D2 have been erosionally truncated.

The Miocene stratigraphic marker is a medium to high amplitude surface which cuts the upper D2 reflectors (Fig. 8 and 11c, d). It is an irregular unconformity surface characterized by a series of multiple NW–SE trending, sinusoidal crosscutting channels (Fig. 11c, d). Individual channels are ~500 m in width with a long axis trending parallel to the slope and hence continental margin. The channels form a ~14 km wide zone in the upper continental slope between ~1 200–1 500 mbsl (Fig. 2). The large extent of crosscutting and overlapping makes it difficult to differentiate between individual channels and their lateral extents. Sequence D3 is the youngest Cenozoic, and therefore uppermost, sequence. It forms a 100–350 m seaward thickening wedge (Fig. 6) characterized by medium to low amplitude, mounded to parallel reflectors. Basal sediments of sequence D3 are mounded in geometry for approximately 80 m whereas earlier sediments were deposited continuously in a parallel manner (Figs. 6 and 7). The youngest stratigraphic marker, the seafloor, is a very high amplitude and thick surface. Slumping, characterized by chaotic internal reflectors, is evident on the present-day seafloor (seen in NW of Fig. 7 and a portion of Fig. 9). The slump scar is ~7 km in width, sitting directly above the region of a large Late Cretaceous syncline (B2 sequence) (Fig. 7).

## 4.2 Structural framework

The sedimentary succession comprises several sedimentary packages displaced by a complex structural framework of faults, as shown in Figs. 5 and 11. Since the seismic volume is heavily faulted (over 500 manually picked), the only faults included in the structural framework in Figs. 5b and 11a were large, first-order faults greater than ~2 km laterally. Faults terminate at either the early Campanian, Oligocene, or Miocene surfaces and originate from Turonian or deeper Albian surfaces depending on their location with respect to the DWFTB system. Upon closer inspection, and the use of the variance seismic attribute, it

is evident that some faults extend past the lower Albian surface for both the compressional (Figs. 6a and 7) and the translational domains (Fig. 8). As explained previously, Fig. 5a shows the variance attribute in TWT at time slice 3 424 ms which was used to enhance the lateral continuity of faults. The position of the variance time slice is shown on the regional sections (Figs. 6, 7 and 8). Using the regional sections (Figs. 6, 7, and 8) and Figs. 5 and 11, the structural framework is described in relation to the compressional and translational domains in the following subsections as the geometry and displacement characteristics of

each domain differ greatly.

### 4.2.1 Compressional domain

DWFTBs occur down-dip in the compressional domain with slip mainly upon the Turonian seaward-dipping surface (Figs. 6, 7 and 9). Imbricate thrust faults detach mostly against the Turonian surface, with some extending to the Albian surface at depth and fewer continuing to greater depths. Thrust faults terminate either just below or at the early Campanian surface within the

365 C1 late drift sequence (Fig. 6). Faults are relatively equally spaced between 1.6–2 km, with average displacements of 250 m (Fig. 6). Thrust faults strike NW–SE and dip between 22–45° NE with generally lower dip angles where they detach from the Turonian or Albian surfaces at depth (Figs. 6 and 11a, b). Thrust sheets are segmented and displaced by extensive oblique-slip normal faults measuring up to ~20 km in length (Figs. 5, 7 and 12a, b). Figure 7 is a distal inline section (see Fig. 5c) showing a few normal, oblique-slip and thrust faults extending past the Turonian to the Albian surfaces at depth. In Fig. 7 the upward

vertical extent of oblique-slip faults terminate along the Maastrichtian to Miocene surfaces while normal faults terminate at between the Campanian (early and late) to Maastrichtian surfaces.

Oblique-slip faults strike NE–SW (Figs. 5 and 11a, c) with average dip-slip offsets of ~80 m (Fig. 7). The strike-slip displacement between each thrust sheet segmented is variable showing mainly sinistral (left-lateral) motion. They dip on average between 40–70° mostly to the NW with the few dipping SE in the south showing right-lateral dextral slip motion

(Figs. 5 and 11a, c). In the compressional domain, the concave upwards, NW-dipping oblique-slip faults display a roughly oval-shaped pattern or geometry in plan section (Figs. 5 and 11a). This central zone forms a very broad anticline in sequence B2 (Turonian to Santonian stratigraphic markers) observed SE of the syncline previously mentioned in Fig. 7 (see Fig. 5a, c for location). Down-dip this central zone is not as heavily faulted as its surroundings but contains smaller ~2 km length normal faults dipping both NW and SE with some displaying overlapping or step-like, en-echelon type geometries (Fig. 5). Up-dip

thrust segmenting oblique-slip faults have the greatest strike-slip offsets (~250 m) towards the translational domain (Fig. 5c). Along-strike, sinistral oblique-slip displacements decrease proximally towards the translational domain compared to the down-dip compressional domain, until only normal dip-slip motion occurs where a few thrust sheets are present (Fig. 5). A deviation from the sinistral motion of displacement is seen S of the study area, as occurring just outside the oval-shaped region in plan section shown in Figs. 5a, c and 11a. Here, the oblique-slip faults dip to the W and SE and show dextral (right-lateral) slip motion in contrast to those in the rest of the study area (Fig. 5). This more structurally complex region has smaller localized sets of fold-and-thrust belts occurring below the kilometre-scale DWFTB system (Figs. 6 and 9). The smaller scale DWFTBs occur within sequence B1, detaching from the Albian surface. The smaller set of secondary DWFTBs are less well-defined and weakly folded with smaller thrust spacings of ~250 m and displacements below the seismic resolution limit compared to those of the overlying kilometre-scale DWFTBs. Similar sets of clear, small-scale DWFTBs are found in other portions of the compressional domain between the Albian and Turonian surfaces.

### 4.2.2 Translational domain

From east to west the translational domain is characterized mainly by normal faults, followed by laterally extensive oblique-slip faults and a few segmented thrust faults in the down-dip direction (Figs. 5, 6 and 11a). Most normal and oblique-slip faults in the proximal translational domain extend from the Albian surface, at depth, and terminate between the late Campanian to Miocene surfaces (Fig. 8). The greatest dip-slip fault displacements of normal and oblique-slip faults occur within sequence B2, between the Turonian to Santonian surfaces for both the compressional and translational domain, with offsets reaching up to 80 m (Figs. 6, 7 and 8). Normal faults observed in the study area have shorter lateral extents in comparison to thrust and oblique-slip faults, reaching up to 13 km (Fig. 5). Like oblique-slip faults, they dip 50–70° on average, with a NE–SW orientation perpendicular to thrust faults of the compressional domain (Figs. 5 and 11a, c). Along individual fault planes, dip angles as low as ~30° may occur at depth, while the higher dip angles occur in stratigraphically higher sequences (Fig. 11 c).

In the north of the study area, laterally extensive NW dipping normal faults (> 5 km) form conjugate pairs with laterally shorter normal SE dipping faults (< 2 km length) (compare Fig. 5 with Fig. 8). The overarching dominant fault directions are evident in Fig. 5b, as compared to those seen in the interpretation of the regional section (Fig. 8) since all faults, regardless of their lateral length, were interpreted in the 2D seismic sections. The centre of the translational domain to the east of the study area then becomes dominated by SE dipping normal faults (Fig. 5b). In the south SW dipping faults dominate (Fig. 5b). Second-order normal faults (those that originate from a large first-order fault) are common in the late drift to early Cenozoic megasequences. These attach to the main faults at acute angles with intersections in megasequence C (Fig. 8). Due to their small vertical and lateral extent (<2 km), they were not included in the structural framework shown in Fig. 5b and 11a given that this study focusses on the regional scale structures affecting basin evolution.

 **5 Discussion**

**5.1 Stratigraphy of the Late Cretaceous SW African margin**

The architectural stratigraphy of the deepwater Orange Basin is explained primarily from regional section Fig. 6 because of its perpendicular orientation to the SW African palaeo- and current coastline which was also parallel to the direction of sediment transportation. Terrigenous sediments observed in the Orange Basin were transported offshore by the Orange and Olifants river systems and their ancestral equivalents (Fig. 1). To understand the full stratigraphic framework of the SW African margin we present our interpretations of the deepwater Orange Basin stratigraphy and discuss what has been observed from previous Orange Basin studies from shallow to deepwater environments.

**5.1.1 Influence of multiple shale detachment surfaces**

Key surfaces in the Late Cretaceous megasequence are the Albian and Turonian stratigraphic markers upon which small-scale and large-scale DWFTB systems are found, respectively. The sequence between these two stratigraphic markers (sequence B1 in this study) is comprised of organic-rich shales as revealed from well log data recovered from DSDP 367 and well A calibrated by Baby et al. (2018) in the Orange Basin. The Albian and Turonian markers are MFSs corresponding to the 14At1 and 15At1 shales (Fig. 4; Brown et al., 1995), upon which gravitational sliding occurs (Fig. 6). These shale detachment surfaces are also proven source rock intervals in the basin and are thus of great significance for hydrocarbon potential (Fig. 4; van der Spuy, 2003). Subtle ramps in the early drift (sequence A) and beginning of the late drift megasequence (sequence B1) have linked these multiple levels of basal slip giving rise to a complex range of geometries both within and above the DWFTB systems (e.g., Fig. 7).

Two important factors controlling gravitational collapse within a basin include the stratigraphy of the margin and thickness of the detachment surface (Rowan et al., 2004; Dalton et al., 2017). In the Orange Basin these factors vary due to the presence and distribution of multiple basal over-pressurized shale detachment surfaces (de Vera et al., 2010; Kuhlmann et al., 2010; Dalton et al., 2017). Once enough overburden material has accumulated in the basin, combined with a lack of cohesion along a weak, sloping detachment surface, gravitational sliding occurs (Rowan et al., 2004; Morley et al., 2011). Rapid sedimentation rates with resultant high internal pore fluid pressures, seismicity, downslope undercutting and tilting glide planes are the main trigger mechanisms responsible for slope failure leading to gravitational collapse (Séranne and Anka, 2005; Rogers and Rau, 2006; Kuhlmann et al., 2010).

Dalton et al. (2017) present a model for the temporal evolution of a gravity collapse system in the Orange Basin containing multiple shale detachment surfaces whereby: 1) gravitational collapse initiates in the translational domain (according to Dalton et al., 2015) upon an initial upper detachment surface, followed by up-dip extension then down-dip compression; 2) once the initial detachment surface can no longer accommodate sliding (e.g., due to thinning of the detachment surface, or possibly a

change in shale overpressures), the gravitational system will lock and deformation will cease; 3) stress and strain is redistributed as extensional faults penetrate to a stratigraphically lower shale detachment surface; 4) once the shales are sufficiently compacted (Dalton et al., 2015), a compressional zone forms down dip upon this lower shale detachment surface; 5) if more shale detachment surfaces exist at depth, even older gravitational collapse systems will form following locking and fault renewal with increasing strain. This model implies a downward rather than upward propagation of faults accounting for the formation of underlying smaller DWFTB systems upon an older shale detachment surface. In this study, however, we propose the smaller, localized folds and thrusts observed below the main, kilometre-scale Late Cretaceous DWFTB system simply represent an older gravitational collapse system in older sediments.

### 5.1.2 The Late Cretaceous DWFTB system

The strongly mounded geometry in sequence B1 reflects basin floor sediments of the translational domain (Fig. 8; Baby et al., 2018). Due to the loss of seismic resolution with depth in very poorly seismically imaged areas, the geometry of basin floor turbidite sediments may be erroneously misinterpreted as smaller scale DWFTBs upon the Albian shale detachment surface. Careful inspection was therefore required to properly differentiate between one type of mass flow or transport system from another, especially within the poorly imaged B1 sequence. Overlying the Turonian shale detachment surface is a much larger scale DWFTB system within late drift sequences B2 and C1, showing that major gravitational collapse began during the Turonian. The folded anticlines of the gravitational system are antiformal as the stratigraphic sequence does not appear to be refolded, and subsequently associated synclines are synformal. The major basal shale detachment surface is the Turonian as shown throughout the seismic volume (Figs. 6, 7 and 8) and recognised throughout the Orange Basin (e.g., **Paton et al.,** 2008; de Vera et al., 2010; Scarselli et al., 2016) while the Albian is minor with smaller, localized fold and thrust faults detaching it in a few areas, e.g., Fig. 6.

The downslope compression of sediment from the up-dip extensional (not imaged) and translational domains thickened and strongly folded sequence B2 during gravitational sliding, with the greatest thickness shown to have occurred at the onset of compression (Figs. 6 and 9). The irregularity and discontinuous nature of the Turonian surface is likely caused by the thinning out of the surface under the weight of thick overlying B2 sequence as it slid seawards with progressive deformation. The large thickness of sequence B2 sequence indicates high sedimentation rates during the Coniacian to early Campanian. In comparison, the overlying sequence C1 is thinner, weakly folded, and has smaller thrust displacements suggesting a syn-kinematic relationship whereby thrusting was still active and contemporaneous with the deposition of these Santonian to early Campanian sediments. In the translational domain the onlap and downlap of sequence C1 upon the Santonian surface (shown well to NE in Fig. 6 and NW of Fig. 7) indicate a localized increase in accommodation space above the translational domain during the downward compression and translation of sediment (Fig. 6). Onlap may also imply lower sedimentation rates relative to deformation which is evidenced by the dramatic thinning of Santonian sediments seawards (Fig. 6). Downslope thrusting is proposed to have ended by the end of the late Campanian as the configuration of reflections in sequence C2 changed from

weakly folded to unfolded, parallel reflections towards the deposition of the upper bounding late Campanian surface shown in Fig. 6. This observation correlates with that of a DWFTB system observed directly north of the study area in the Namibian Orange Basin (Fig. 1) where the estimated end of gravitational deformation is also postulated within Campanian sediments (~83 Ma) from the analysis of growth stratal patterns (Fig. 1; de Vera et al., 2010). The early Campanian sediments, sequence C2, are interpreted to have been deposited under normal to moderate rates of sediment supply as reflections are conformable and continuous and the proximal to distal thickness of the sequence does not change considerably. Sequence C3's late Campanian to Maastrichtian sediments thin seawards with the distal decrease in sediment supply. The Santonian, early Campanian, late Campanian, and Maastrichtian surfaces correspond to the 16Dt, 17At1, 18 At1 and 22At1 stratigraphic markers (erosional or non-depositional surfaces) (Fig. 4; Brown et al., 1995; PASA, 2017).

The Early Cretaceous post-rift megasequence is retrogradational reflecting the change from subaerial conditions during the emplacement of the SDRs (syn-rift succession) to an open marine environment with a narrow oceanic basin between the late Hauterivian and Albian (de Vera et al., 2010; Baby et al., 2018). As the margin continued to deepen with retrogradation during the late Albian to early Turonian, a well-defined depositional profile with shelf, slope and basinal elements began to form (Light et al., 1993; Baby et al., 2018). The Late Cretaceous post-rift megasequence is comprised of aggradational to progradational packages reflecting increased sedimentation rates, and a decrease in subsidence from the early Turonian to late Maastrichtian (~93.5–66 Ma) attributed to the late Campanian sea-level fall with major uplift and seaward tilting of the margin which eroded and transported sediments to the Orange Basin as supported by thermochronometric data (Paton et al., 2008; Granado et al., 2009; de Vera et al., 2010; Kuhlmann et al., 2010; Hirsch et al., 2010; Baby et al., 2020). This period in time records the greatest flux in sedimentation offshore the SW African margin with the Orange Basin showing lower rates and volumes between 81 and 66 Ma (Baby et al., 2020). The Maastrichtian stratigraphic marker (22At1) is a maximum regressive surface and transgressive ravinement surface as it separates the Cenozoic retrogradational megasequence from the underlying Late Cretaceous progradational megasequence (Fig. 4, Baby et al., 2018).

## 5.2 Structure of the Late Cretaceous DWFTB system

The kinematics, geometry and displacement characteristics of structures seen within the translational and compressional domains imaged in this study differs greatly from each other. We discuss the variability of the two domains imaged to assess the strato-structural evolution of the Orange Basin from the Late Cretaceous DWFTB system to overlying Cenozoic deposits. The lateral and vertical change in structures between the translational and compressional domains is seen through progressive deformation and sedimentation.

### 5.2.1 Compressional and transitional tectonics

The structural framework reveals that the transition between translation and compressional tectonics is gradational with normal and oblique-slip faults dominating proximally, and thrust and oblique-slip faults dominating distally (Figs. 5b, 11a). Landward-

dipping imbricate thrust faults with seaward-verging folds are recognised in the compressional domain (Figs. 5, 6, 9 and 11a). The NW-SE orientated thrust faults dip gently landwards, detaching the Turonian shale detachment surface and terminating at the early Campanian surface (Fig. 6). A few thrust faults reach the Albian shale detachment surface at depth and even extend past into stratigraphically lower sequences (Figs. 6 and 8). This shows that renewed faulting caused by gravitational deformation in the Late Cretaceous DWFTB system occurred preferentially along the older pre-existing planes of weakness. Proximal normal and oblique-slip faults extend into stratigraphically younger Cenozoic sediments and higher surfaces; most detach the Albian shale detachment surface and terminate at either the late Campanian, Oligocene, or Miocene surfaces (e.g., Fig. 8). Since thrusts of the down-dip compressional domain terminate at the early Campanian surfaces, faults in the proximal domain likewise probably originally terminated at here. With the subsequent deposition of overlying sediments, however, faults were renewed to propagate upwards into stratigraphically younger sequences, accommodating further increments in strain. This supports the idea that once formed, a fault or fracture will always represent a mechanical zone of weakness to be exploited (Viola et al., 2012).

Oblique-slip faults, formed from the combination of tension and shearing during the down-dip translational of sediment, crosscut thrust sheets orthogonally and thus post-date them. We interpret en-echelon type and stair-stepped geometries of the small normal faults observed in the centre of the compressional domain imaged in Figs. 5 and 11a to be indicative of lower rates of tensional shear. This is because they are relatively small (~2 km in length laterally) in comparison to the kilometre-scale oblique-slip faults they are bound within which indicate higher rates of tensional shear. The outer oblique-slip faults accounting for the oval geometry observed in plan view in Fig. 5 are shown to have rotated the Turonian to Santonian (sequence B2) sediments (Fig. 7). The observed strike-slip offset for each oblique-slip fault represents the hanging wall and footwall blocks of each thrust sheet since hanging wall and footwalls have the opposite sense of motion (Benesh et al., 2014). The downslope vertical shortening of oblique-slip faults from the translational to compressional domains is shown from regional section Fig. 8 (translational) to Fig. 7 (compressional) and furthermore illustrated in Fig. 11c. Over and above the decrease in basin thickness from proximal to distal settings, explained previously in Section 5.1, we attribute this shortening to the decrease in tension and shearing during the downslope translation of sediment. Other DWFTB systems exhibiting similar behaviour to that observed in this study are found in the Niger Delta containing multiple shale detachment surfaces and oblique-slip faults which segment thrust sheets along strike (Rowan et al., 2004; Benesh et al., 2014).

### 5.2.3 Extensional tectonics

Although the extensional domain is not imaged in this study, other studies of DWFTB systems in the Orange Basin may be used to address the structural framework in the extensional domain, e.g., Paton et al. (2008), de Vera et al. (2010), Scarselli et al. (2016). Down-dip compression is known to be linked to up-dip extension domain shown from full DWFTB systems in the Orange Basin (Fig. 3; de Vera et al., 2010). Listric growth faults dip seaward in the extensional domain while in the compressional domain thrust faults dip landward and thus display an arcuate geometry in 3D as explained by de Vera et al.

(2010) and Scarselli et al. (2016). The same geometry may be deduced in this study which lies directly south of the DWFTB system observed by de Vera et al., (2010) and shown in Fig. 3. The unimaged extensional domain of the study area likely lies along the continental shelf.

### 5.2.4 Geometry of structural framework

Looking at the overall structural framework, the dominant azimuthal dip direction in a clockwise direction changes NE to NW
to SE to SW dipping faults (Fig. 5b). Normal and oblique-slip faults dip steeply with NE–SW oriented strikes in contrast to the gently dipping orthogonal NW–SE thrusts, formed by down-dip sedimentary contraction. Most structural lineaments, and the elongation of plutons and igneous intrusions along the SW African margin are orientated parallel to the regional NW–SE foliation trend of the margin, with the exception of a few complexly faulted regions (Wildman et al., 2015). Thrust faults imaged in this Orange Basin study follow this same NW–SE predominant trend. We deduce the same for the up-dip extensional
domain based on other DWFTB systems in the region (e.g., Fig. 3; de Vera et al., 2010; Scarselli et al., 2016). Normal and oblique-slip faults, however, trend orthogonal to the thrust faults (Figs. 5, 11a) and regional foliation of the margin (Fig. 1). Notably, oblique-slip faults are orientated parallel to the deep-seated, underlying Rio Grande and Aghulhas-Falkland fracture zones which accommodated dextral slip motion during rifting of the SW African margin (Fig. 1; Light et al., 1993; Séranne and Anka, 2005). Most oblique-slip faults in the study, however, segment thrust sheets in a sinistral mode of motion apart from
the region in the south (Fig. 5a, c). The left-lateral sinistral sense of motion (Fig. 5c) may indicate gravitational collapse initiated in the south and moved northwards with each subsequent thrust sheet moving seawards. The oblique-slip faults acted as planes for the lateral displacement of each thrust sheet.

In the southern region of the studied volume, the kinematics and geometry of the structural framework differs. Here, right-lateral dextral slip motion of thrust sheets is observed, which is associated with opposite SW dipping normal and oblique-slip
faults (Fig. 5). The change in geometry and sense of movement may represent the outer oblique-slip faults of another gravitational collapse system south of the studied volume. This is shown and described by Scarselli et al., (2016) N of the present study between 350–1 300 m water depths along the upper continental slope of Namibia. Both 2D and 3D seismic reflection data reveal a fuller DWFTB system including the up-dip extensional domain with seaward-dipping listric growth faults and connected sidewall faults rotating Late Cretaceous sediments to form rollover anticlines in gravitational collapse
systems. The outer oblique-slip faults described centrally in this study are analogous to the sidewall faults they described for each megaslide complex showing lateral downslope motion with gravitational collapse. The megaslides occur adjacent to each other with opposite N and S dipping sidewall faults, forming horst structures separating each megaslide. This too describes and accounts for the change in geometry and kinematics of the structural framework in the S of the present study. Notably, however, our main outer oblique-slip faults predominantly dip in the same NW or N direction. Only the northern outer oblique-
slip fault is shown in Fig. 7 (see location of regional line in Fig. 5b) which is where a Late Cretaceous syncline and seafloor slump scar is found.

### 5.2.5 Translational domain implications

There are two different schools of thought in what the translational domain represents:

1. A zone caused from the central shift in contact between extensional and compressional tectonics containing overprinted features of both (Butler and Paton, 2010; de Vera et al., 2010). This was proposed due to the crosscutting relationships observed in the distal parts of the extensional domain.
2. A fixed package of mostly undeformed rock representing a short wave-length change from extensional to compressional tectonics (Bilotti et al., 2005; Corredor et al., 2005; Krueger and Gilbert, 2009; Dalton et al., 2017).

In previous Orange Basin studies, the transitional/translational domain was interpreted as a relatively narrow region (~10 km in length perpendicular to the slope) considering the massive extent of a complete (roughly ~220 km) DWFTB system (Corredor et al., 2005; Krueger and Gilbert, 2009; de Vera et al., 2010; Dalton et al., 2017). In this study, the true extent of the translational domain cannot be constrained since the linked up-dip extensional domain is not imaged. Nevertheless, what is observed is that the translational domain reaches and is greater than ~20 km in width, as observed in the north of the studied volume. We therefore propose a third model for what the translational domain represents; a zone displaying overprinted features of both extensional and compressional tectonics, depicting a long wavelength change between the up-dip and down-dip domains with the downslope translation of sediment accommodated by extensive oblique-slip faults.

Overlapping compressional and extensional tectonics may possibly be described in terms of the misbalance in strain observed in other Orange Basin studies. In linked DWFTB systems, the amount of extension should equal the amount of compression; this is however not the case as reflected through the approximate 5% missing strain observed in favour of extension accounted for in second order, compressional structures within the extensional domain itself (Butler and Paton, 2010; de Vera et al., 2010; Dalton et al., 2015). The lack of a down-dip contractional domain in the outer reaches of some of these systems, as noted by Dalton et al. (2015) for the Orange Basin, is proof that the 5% missing strain is accommodated for internally. In like manner, the overlapping and crosscutting relationships observed in the transitional domain is the system redistributing strain internally. In response to up-dip extension, down-dip compression occurs and with progressive deformation oblique-slip faults act as lateral ramps to accommodate strain with the growth formation of each subsequent thrust sheet. Further small-scale extension occurs within each individual thrust sheet to accommodate strain with gravitational collapse as evidenced by the en-echelon and step-type normal faults in the compressional domain.

### 5.3 Cenozoic stratigraphy of the SW African margin

### 5.3.1 Canyon-channel systems

One of two major features recognised in the Cenozoic megasequence is a large SE to NW orientated canyon which defines the Oligocene erosive surface (Figs. 8, 10a, b). The canyon eroded the underlying early Oligocene (sequence C3 which downlaps

the Maastrichtian surface) to late Campanian (sequence D1) sediments (Fig. 8). Sequence D2, the sedimentary infill of the canyon, onlaps against the main Oligocene erosional surface and ensuing erosional surfaces. The Oligocene canyon is interpreted as having been formed by the erosive action of a turbidity current with downslope canyoning possibly deflected by a strong north-ward flowing oceanic current since its orientation is not perfectly perpendicular to the continental slope. Turbidity currents are episodic sediment gravity-driven flows that always flow down-slope, unless the local morphology (such as the presence of marginal troughs) forced slope-parallel flow (Shanmugam, 2008; Shumaker et al., 2016). The infill of the canyon (sequence D2) displays some chaotic internal reflections possibly diagnostic of turbidite deposits (Fig. 8).

The second major feature in the Cenozoic are the extensive margin-parallel sinusoidal and overlapping channels, defining the Miocene erosive surface (Figs. 8, 10b, c). These erosional features are interpreted to have been formed by subcircular water and/or sediment motions flowing parallel to the margin (i.e., erosive deep-water bottom currents) due to their margin-parallel orientation. Supporting this interpretation is the fact the channels are confined between specific isobaths along the palaeo-slope with the base of the overlying sequence D3 displaying mounded geometries indicative of mass flow (Figs. 8, 10c, d). This contrasts with the very roughly slope-perpendicular Oligocene canyon proposed to have been formed by a turbidity current (Fig. 10). A bottom current that remains active over prolonged periods of time (i.e., millions of years) will affect sedimentation on the ocean floor, from the winnowing of fine-grained sediments to large-scale erosion and deposition of coarse-grained sediments (Shanmugam, 2008; Rebesco et al., 2008). Strong bottom currents preserve slope-parallel erosional features reflecting the contoured flow of either a thermohaline-, wind-, or tidal-driven (baroclinic currents) deep-water bottom current (Shanmugam, 2008). Temperature and/or salinity differences in bodies of water along the continental slope cause vertically stratified water layers of different densities resulting in baroclinic/geostrophic flow.

Chaotic internal reflections of sequence D3 and the depression of the seafloor characterize a major slump scar reflecting margin instability in Fig. 7. Internally, slumps are highly deformed due to their rotational movement upon a concave-upward glide plane (Shanmugam, 2017). The slump scar is located directly above a syncline within Late Cretaceous sediments (sequence B1) showing the underlying structural control of the Late Cretaceous DWFTB compressional domain. Faults from the gravitational system, however, are not shown to reach the seafloor, but rather terminate along the Miocene stratigraphic marker unless their displacements are below the seismic resolution limit (Fig. 7). Notably, both the Oligocene and most of the Miocene canyon-channel systems are located above the transitional domain. The position of the Oligocene canyon initially appears to be fault controlled since it occurs directly above two opposite dipping faults, forming a horst and graben feature in Fig. 8. In 3D however, (see Fig. 5) both the Oligocene canyon and Miocene channels do not appear to be influenced by the underlying tectonics of the Late Cretaceous since they do not follow the trend of underlying faults. Rather, the vertical propagation of faults in the transitional domain was hindered by the mass flow deposits overlying the Oligocene and Miocene stratigraphic markers.

### 5.3.2 Role of tectonics and oceanographic circulation on Cenozoic sedimentation

The SW African margin is characterised mainly by retrogradational sequences in the Cenozoic (e.g., Baby et al., 2018) with elevated sedimentation rates occurring in the Oligocene between ~30–25 Ma (Baby et al., 2020). The Oligocene stratigraphic marker corresponds to a major event recognised in literature and seen throughout the African shelf (Fig. 4; Siesser and Dingle, 1981; Miller et al., 1995; Séranne and Anka, 2005). It was caused by margin uplift resulting in a relative sea-level fall (possibly beyond the shelf break which is a topic of controversy in literature) and is evidenced by; tilting topsets in the Walvis Basin, and a 350 m difference in elevation between the late Eocene and Oligocene shorelines of the northern Orange Basin at ~30 Ma, as observed in the shallower reaches of the SW African margin (Hirsch et al., 2010; Baby et al., 2018). Along the coast, well data from Saldanha Bay record sea levels as low as -100 m during the Oligocene (Roberts et al., 2017). The early Miocene corresponds to a sea-level lowstand caused by tectonic uplift attributed to the African superswell (Fig. 3; Séranne and Anka, 2005; Wigley and Compton, 2006; Hirsch et al., 2010). Following this, a major decrease in the supply of siliciclastic sediment is recorded along the SW African margin at ~11 Ma, accounting for the Miocene stratigraphic marker (Figs. 6, 7, 8 and 10; Baby et al., 2018). The decrease in sediment supply in the Miocene is attributed to the aridification of the Namibian margin at ~17–15 Ma and the lack of margin uplift thus reducing river flooding (e.g., Siesser, 1980). Controls other than tectonics and the morphology of the margin need to be considered to fully understand the stratigraphy of the Cenozoic megasequence, particularly for the mid-Miocene till present as no major uplift event is recognized for the SW African margin during this time (e.g., Siesser and Dingle, 1981).

The Miocene sinuisoidal channels in this study were formed by the interaction of parallel flowing bottom currents together with upwelling-induced erosion in the south which both began at around the same time (see Section 2.4 for a full overview; Weigelt and Uenzelmann-Neben, 2004, 2007a). Based on the sequence stratigraphic interpretation of sediments in the Cape Basin, Weigelt and Uenzelmann-Neben (2004, 2007b) propose upwelling to have initiated at around ~14 Ma up until ~1.5 Ma, and therefore earlier than the ~11 Ma suggested by Diester-Haass et al. (2004) and Rommerskirchen et al. (2011). Together with the onset of the Benguela upwelling system (BUS), the dramatic change in late Miocene to Plio-Pleistocene sedimentation is attributed to the strong influence of slope-parallel deep and bottom water currents formed at ~10 Ma when terrigeneous sediment sourced from the Orange and Olifants river systems was negligible (Weigelt and Uenzelmann-Neben, 2004, 2007a). The bottom and deepwater currents offshore western South Africa include the upper warm northward flowing Antarctic Intermediate Water (AAIW) surface waters down to ~1 500 m water depths; the North Atlantic Deepwater (NADW) between ~1 500 to 4 000 m water depths; and the deep Antarctic Bottomwater (AABW) beyond 4 000 m water depths (Weigelt and Uenzelmann-Neben, 2004). Since the position of the channels occurs along the upper continental slope between ~1 200 m to 1 500 m, the specific bottom water currents responsible for them are the upper northbound AAIW and deeper southbound NADW. The wide ~14 km wide zone of sinusoidal scouring along the Miocene stratigraphic marker (Fig. 10c, d) therefore marks the intersection of these concurrent flows. The present-day flow of bottom currents is still strong enough to erode the

660 slope as seen off Cape Columbine (South Africa) with pole-ward current speeds of 16.1 cms$^{-1}$ (Compton and Wiltshire, 2009). The slope-parallel channels indicate that current speeds were even greater in the Miocene, strong enough to erode the slope and occurring over a prolonged period with their preservation linked to the lack of terrigenous input from the onshore Olifants and Orange rivers.

Along the outer shelf of northern Namibia, Hopkins and Cartwright (2021) similarly identified a closed ~40 by 50 km kidney
shaped depression that eroded late Miocene sediments. The feature was likely formed by the erosive action of bottom currents associated with the intensification of the late Miocene BUS as evidenced from the orientation of its entry point and the reflection geometry of infilling sediment aligned parallel to the margin. Other possible mechanisms given for the feature included: 1) sea level fall with margin uplift, and 2) slumping or gravitational collapse; both of which were rejected since there is no evidence for a dramatic sea level fall recorded during this time (e.g., Siesser and Dingle, 1981), and sediment has not
been remobilized. Furthermore, no visible channel feeds into the feature and no faults are found beneath it. The configuration of infilling sediments are downlapping clinoforms interpreted as diagnostic of contourites. Contourites are indicative of high sedimentation rates and many of their depositional systems have been discovered in major oceans worldwide whose bottom currents are shown to have largely been initiated around the Eocene-Oligocene boundary at ~32 Ma, then later reactivated, or newly formed in the Miocene, reflecting the global change in oceanic circulation (Rebesco et al., 2014).

Compared to the western South African margin, the Namibian margin has thicker Miocene to Pliocene sedimentary sequences, gradually thickening northwards which is attributed to the dramatic northerly intensification of the BUS between both SW African margins (Weigelt and Uenzelmann-Neben, 2007a). We relate the reworked sediments overlying the Miocene stratigraphic marker (sequence D3) to sediment drifts transported by the erosive action of the BUS, AAIW and NADW (Figs. 6 and 8). In Weigelt and Uenzelmann-Neben (2004), the interaction of the AAIW and NADW is shown to have formed slumps
in the late Cenozoic succession of the Cape Basin. Likewise, in this study, the seafloor slump scar observed in Fig. 7 may possibly have been triggered by the AAIW and NADW, causing mass movement above the weakest zone; directly above a syncline from the Late Cretaceous DWFTB system with fault reactivation. Although faults extending from the Albian shale detachment surface do not appear to reach the seafloor slump scar, it is possible that fault displacements in the younger Cenozoic sediments are simply below the seismic resolution limit. Elsewhere along the western South African margin many
Cenozoic to seafloor slumps have been identified strongly influenced by the underlying structural framework of DWFTB systems (e.g., Paton et al., 2008; Hirsch et al., 2010).

**5.4 Strato–structural evolutionary model of the deepwater Orange Basin**

Using the stratigraphic and structural observations made in the present study, combined with previous literature on the tectonics
and stratigraphy of the Orange Basin (e.g., Weigelt and Uenzelmann-Neben, 2004; 2007a; Séranne and Anka, 2005; Paton et

al., 2008; Granado et al., 2009; Wigley and Compton, 2006; de Vera et al., 2010; Hirsch et al., 2010; Kuhlmann et al., 2010; Scarselli et al., 2016; Baby et al., 2018), we present a model in Fig. 12 illustrating the temporal evolution of the deep-water Orange Basin. Late Cretaceous ages were assigned based on the major unconformities shown in Fig. 4 (Brown et al., 1995; PASA, 2017). Cenozoic ages assigned and processes are based on the major unconformities observed offshore SW Africa formed by tectonic uplift (Oligocene) and oceanic currents (Miocene onwards) (Weigelt and Uenzelmann-Neben, 2004, 2007a, 2007b; Baby et al., 2018). The following sequence of events is proposed from the formation of a Late Cretaceous DWFTB system to overlying Cenozoic canyon-channel systems and slumps in the deepwater Orange Basin:

a) **~103–93 Ma (Albian–Turonian):**
- Following the full opening of the Atlantic Ocean (Fig. 4), well-defined shelf, slope and abyssal plain environments developed along the margin, and gravitational collapse formed small-scale folds and thrusts detaching the Albian shales.

b) **~93–85 Ma (Turonian–Santonian):**
- With margin uplift, listric normal faulting in the up-dip extensional domain led to the formation of fold-and-thrust belts in the down-dip compressional domain with faults detaching the Turonian shale detachment surface. Thrusting was accommodated by perpendicularly orientated oblique-slip faults extending from the translational domain.

c) **~85–77.5 Ma (Santonian–late Campanian):**
- With progressive sedimentation, oblique-slip faults continued to act as lateral ramps for thrust sheets to redistribute strain caused by continued down-dip compression and deformation.

d) **~30–25 Ma (Oligocene):**
- A major sea-level fall following margin uplift led to the formation of a large Oligocene canyon as sediments were eroded by a downslope turbidity current (Figs. 8 and 10a, b).

e) **~12–10 Ma (Miocene):**
- Following margin uplift, multiple crosscutting and sinusoidal channels in the Miocene formed by the slope-parallel erosive action of the concurrent NADW and AAIW ocean currents together with intensification of the BUS (Figs. 6, 8 and 10c, d).

f) **~12 Ma–0 Ma (Miocene–Present):**
- Slope instability triggered by ocean currents formed Cenozoic to present-day seafloor slumps (Fig. 7).

## 6 Conclusions

In this study, we provide insight into the kinematics, geometry and displacement characteristics of structures observed within and above the compressional and translational domains of a Late Cretaceous DWFTB system in the Orange Basin using 3D

seismic reflection data. These insights permitted a greater understanding into the mechanisms responsible for deformation, sedimentation and accommodation following gravitational collapse in the Late Cretaceous to Cenozoic megasequences of the basin. Although no two DWFTB systems may exactly be alike, the observations drawn add to the limited understanding of what occurs in the translational domains of passive margin settings. The following major conclusions have been drawn:

1) **The translational domain of is a zone containing overprinted features of both the extensional and compressional tectonics.**
   - The downslope translation of sediment is accommodated by oblique-slip faulting extending from the translational to the down-dip compressional domains segmenting thrust sheets along-strike orthogonally. The difference and changes in slip along each oblique-slip fault reflects the way in which differential movement is accommodated for between the footwall and hanging wall blocks of the thrust sheets.

2) **The morphology of the margin (i.e., slope steepness) and sedimentation rates controlled structural and depositional elements of the Late Cretaceous Orange Basin.**
   - The Cenozoic canyon-channel systems occur mainly above the Late Cretaceous DWFTB system's translational domain. Normal and oblique-slip fault propagation from the translational domain is hindered by mass flow deposits of the Oligocene and Miocene shown by faults terminating at their stratigraphic markers.

3) **Mid Cenozoic sedimentation and canyon-channel systems were controlled by slope-parallel ocean currents.**
   - The orientation of the Oligocene canyon, formed with sea-level fall, is slightly deflected to the N possibly reflecting the initiation of a northwards flowing current.
   - During the mid-Miocene slope-parallel oceanic currents, together with the lack of sediment supply were the main factors controlling sedimentation patterns with the Miocene erosional features, representing the zone where the northward flowing AAIW and southward flowing NADW intersected.

## 7 Author contributions

NGM conducted the research and wrote the manuscript draft; MSDM and ZJ supervised the main author's research; MSDM and ZJ and JEB reviewed and edited the manuscript.

## 8 Competing interests

The authors declare that they have no conflict of interest.

## 9 Acknowledgements

We would like to thank the National Research Foundation of South Africa (Grant UID: 130186) and the Council for Geoscience for funding the first author's research. Our thanks are also extended to Shell South Africa for providing the 3D reflection

seismic data, and Schlumberger for the Petrel software and support. We are furthermore grateful for the insights Vuyolwethu Mahlalela provided, and to our colleagues and friends from the Wits Seismic Research Centre for the scientific discussions had and invaluable inputs provided. The authors would also like to thank 2 anonymous reviewers and Prof. Chris Elders (Curtin University) for their constructive comments which have greatly improved this manuscript. The editorial team of EGUsphere

is also thanked for editorial handling.

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

**Table 1: Acquisition parameters of the seismic survey (Kramer and Heck, 2014).**

| Recording | |
|---|---|
| *Recording format* | SEG-D |
| *Record length* | 7168 ms |
| *Recording filter delay* | None |
| *Sample rate* | 2 ms |
| *Low cut filter hydrophone, slope* | 4.4 Hz, 12 dB / octave |
| *Low cut filter geophone, slope* | Not applicable |
| *High cut filter both, slope* | 214 Hz, 341 dB / octave |
| **Source** | |
| *Source type* | Dual source |
| *Number of source arrays* | 2 |
| *Number of sub-arrays* | 3 |
| *Shot point interval* | 25 m (flip/flop) |
| *Array separation* | 100 m |
| *Array length* | 15 m |
| *Source volume* | 4100 cu in |
| *Number of airguns / arrays* | 30 |
| *Operating pressure* | 2000 psi |
| *Source depth* | 8 m |
| *Nominal CMP fold* | 80 |
| **Sercel SSAS Sentinel, Sercel Seal-428** | |
| *Number of streamers* | 8 |
| *Group interval* | 12.5 m |
| *Group length* | 12.5 m |
| *Number of hydrophones / groups* | 1 |
| *Number of geophones / groups* | 1 |
| *Streamer length* | 7950 m |
| *Streamer separation* | 200 m |
| *Number of groups / streamers* | 636 |
| *Streamer depth* | 10–15 m linear slant |
| *Nearest offset* | 222 m |


**Table 2: Processing workflow of the seismic survey (Kramer and Heck, 2014).**

| Dolphin Geophysical Polar Duchess team | |
|---|---|
| *1* | Conversion from SEG-D and navigation merge |
| *2* | Output to SEG-Y |
| **Global Processing team** | |
| *3* | Conversion from SEG-Y to Shell's proprietary software (SIPMAP) format |
| *4* | Spherical spreading correction |
| *5* | Despike |
| *6* | Swell noise attenuation |
| *7* | Resample to 4 ms |
| *8* | Denoise |
| *9* | Linear noise attenuation |
| *10* | Deghosting |
| *11* | Seismic interference attenuation |
| *12* | Zero phasing |
| *13* | 2D surface rendered multiple elimination (SRME) prediction |
| *14* | 3D SRME prediction |
| *15* | LSQ matched subtraction |
| *16* | Multiplicity scaling for Kirchhoff migration |
| *17* | Phase deabsorption |
| *18* | Residual moveout (RMO) analysis |
| *19* | RMO velocity model inversion and anisotropy scanning |
| *20* | Pre-migration signal enhancement |
| *21* | Kirchhoff prestack migration |
| *22* | Residual radon demultiplex |
| *23* | Amplitude deabsorption |
| *24* | RMO correction |
| *25* | (Angle) stack |
| *26* | Time variant scaling (additional output volumes) |
| *27* | Archiving |

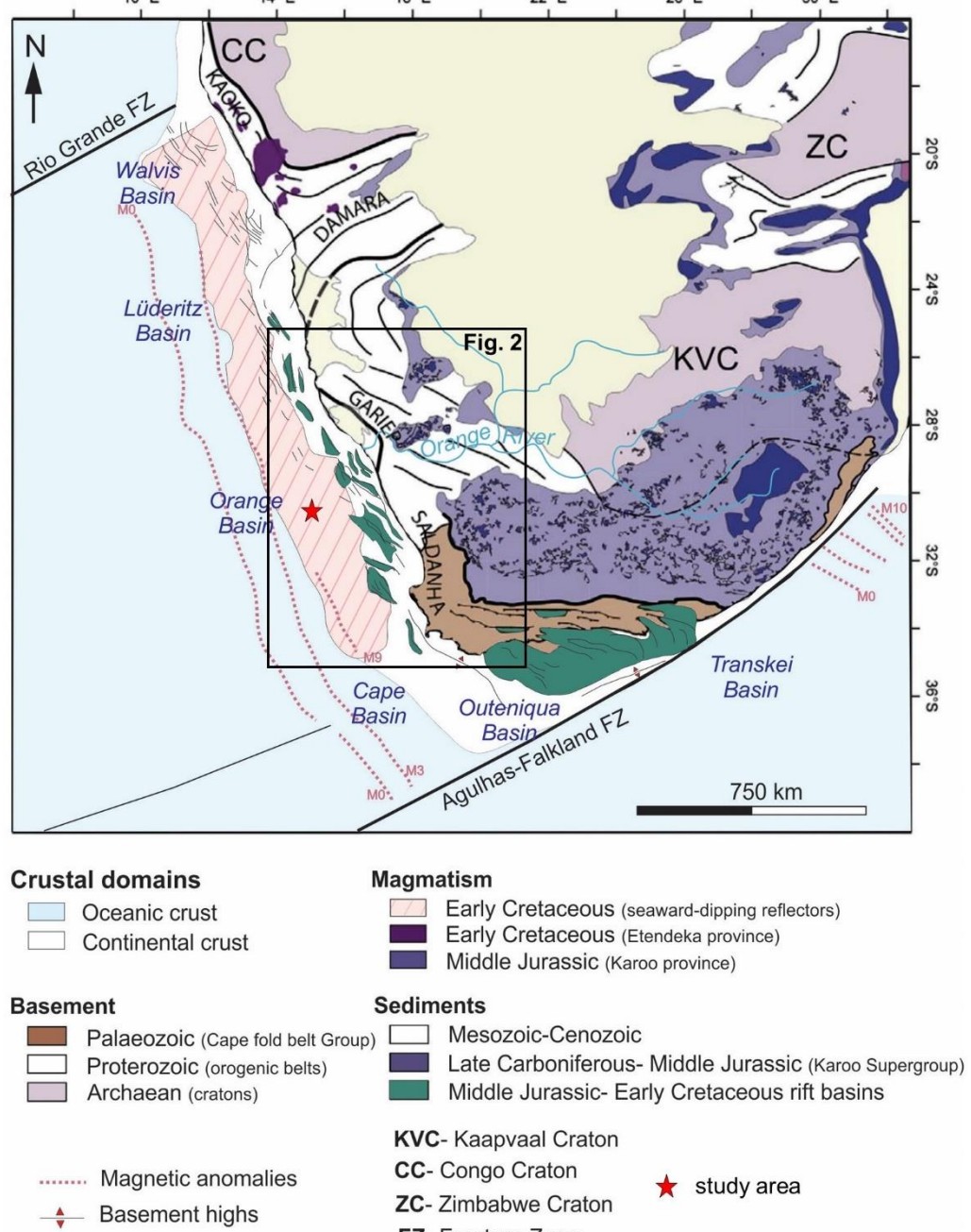


**Figure 1: Tectonic structural framework showing the crustal components of southern Africa and major basinal depocentres offshore the SW African margin (adapted from Baby et al., 2018).**

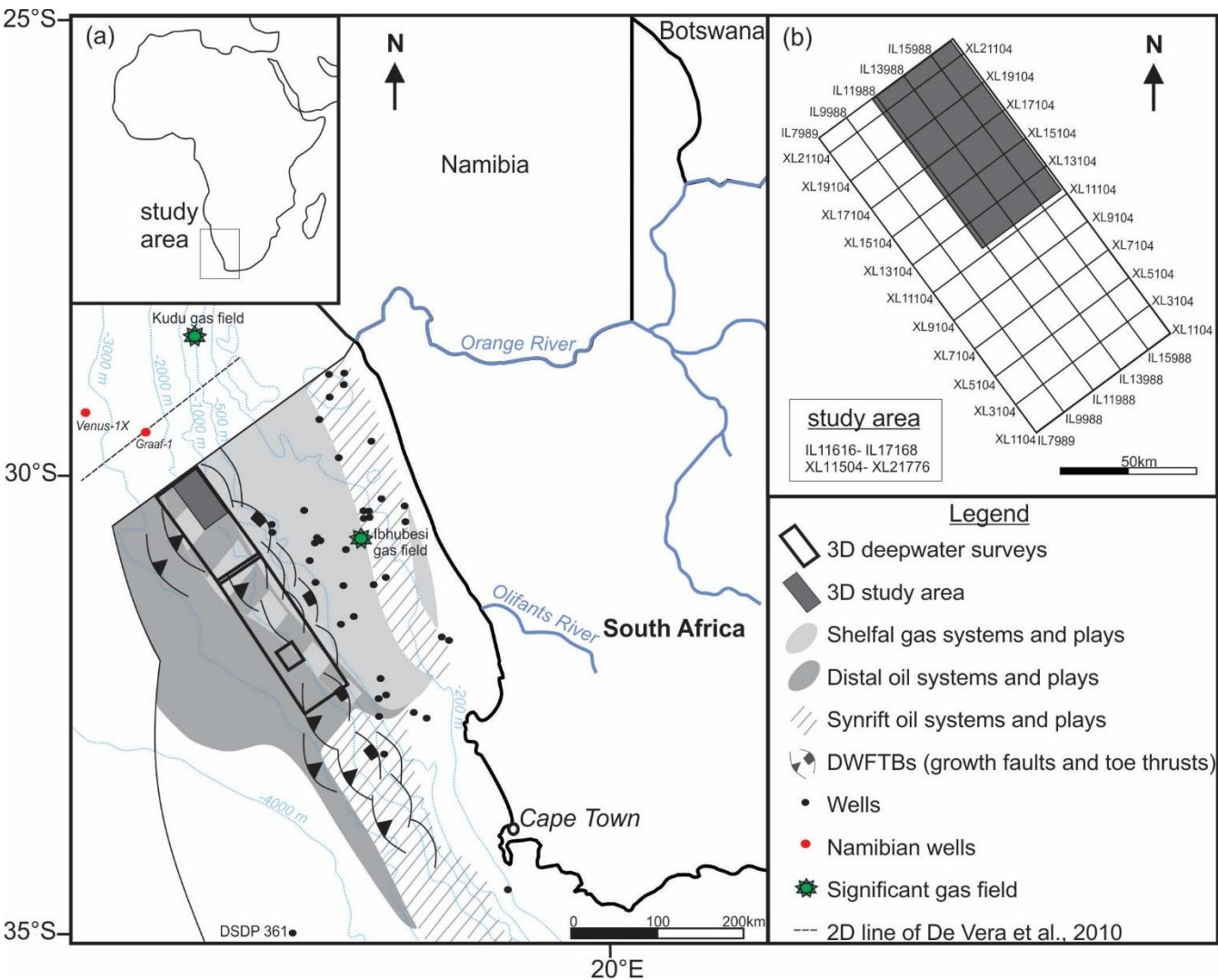


**Figure 2: Simplified map of the Orange Basin study area offshore the SW African coastline. (a) Known and predicted petroleum systems and plays, and the position of gravitational structures and wells within the South African licensing area (adapted from Jungslager, 1999). (b) Location of the present study within the full 3D seismic survey.**

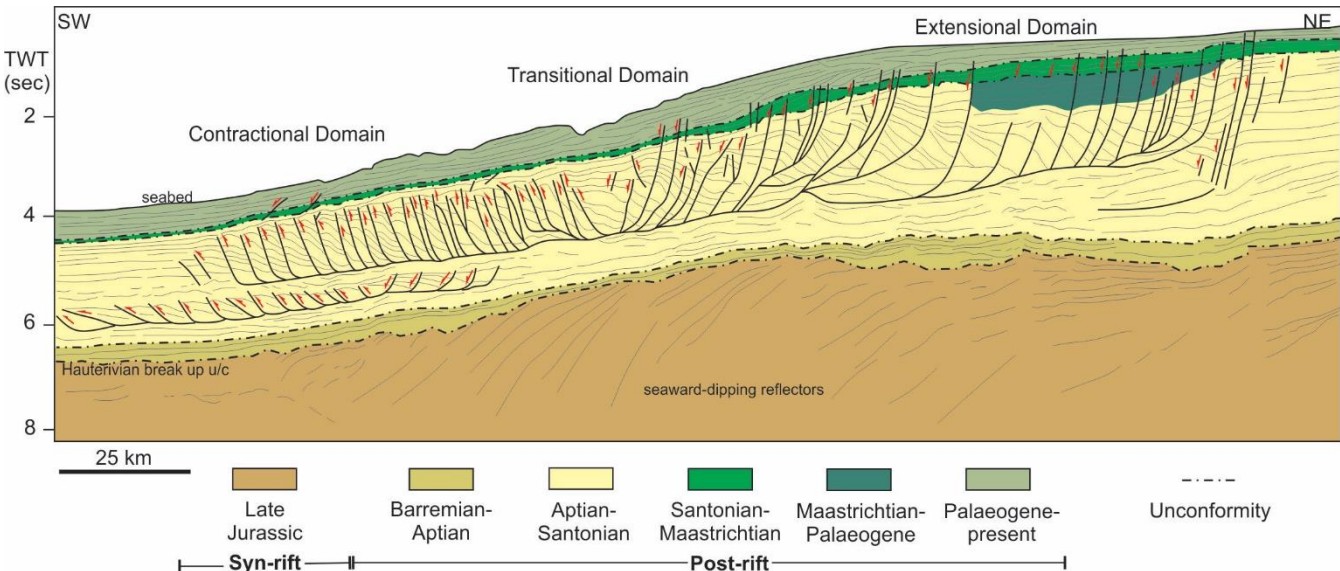

**Figure 3: 2D profile of the Orange Basin showing structures with the up-dip extensional, central transitional (or translational) and down-dip compressional domains of a Cretaceous DWFTB system upon the Late Jurassic syn-rift sequence (De Vera et al., 2010).**

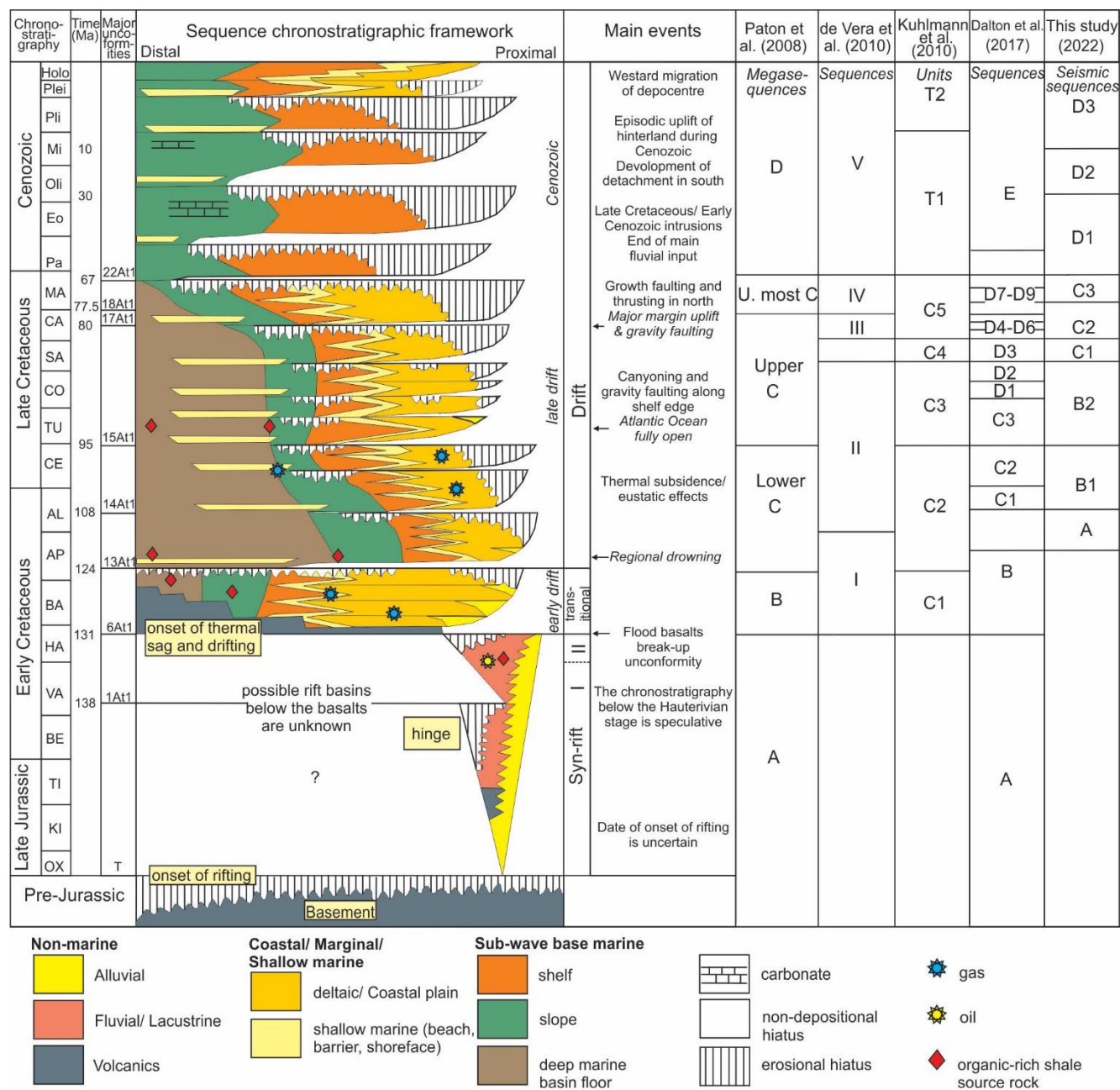

**Figure 4: Tectonochronostratigraphic chart of the Orange Basin showing the major depositional sequences, their bounding surfaces, and the evolution of the margin through fluctuations in global and local sea level (Brown et al., 1994; PASA, 2017).**

**Table 3 Diagnostic attributes of the main sequence stratigraphic surfaces in deepwater marine settings (Catuneanu, 2006).**

| Stratigraphic surface | Description | Nature of contact | Marine Facies | | Depositional trends | Stratigraphic surface |
|---|---|---|---|---|---|---|
| | | | below | above | | |
| **Maximum regressive surface (MRS)** | Marks change from shoreline regression to transgression; replaced by MFS in distal marine | Conformable (few exceptions) | Fining upward in deep water; Coarsening-upward in shallow water | Fining-upward | **Above:** transgression **Below:** normal regression | **Above:** marine onlap **Surface:** onlap, downlap |
| **Maximum flooding surface (MFS)** | Marks end of shoreline transgression | Conformable or scoured | Fining-upward | Coarsening-upward | **Above:** normal regression **Below:** transgression | **Below:** truncation **Above:** downlap **Surface:** onlap, downlap |
| **Correlative conformity** | Reflects the paleo-seafloor at the end of forced regression; correlates with the subaerial unconformity | Conformable | Coarsening-upward | Coarsening-upward on shelf | **Above:** normal regression **Below:** forced regression | **Above:** downlap **Surface:** downlap |

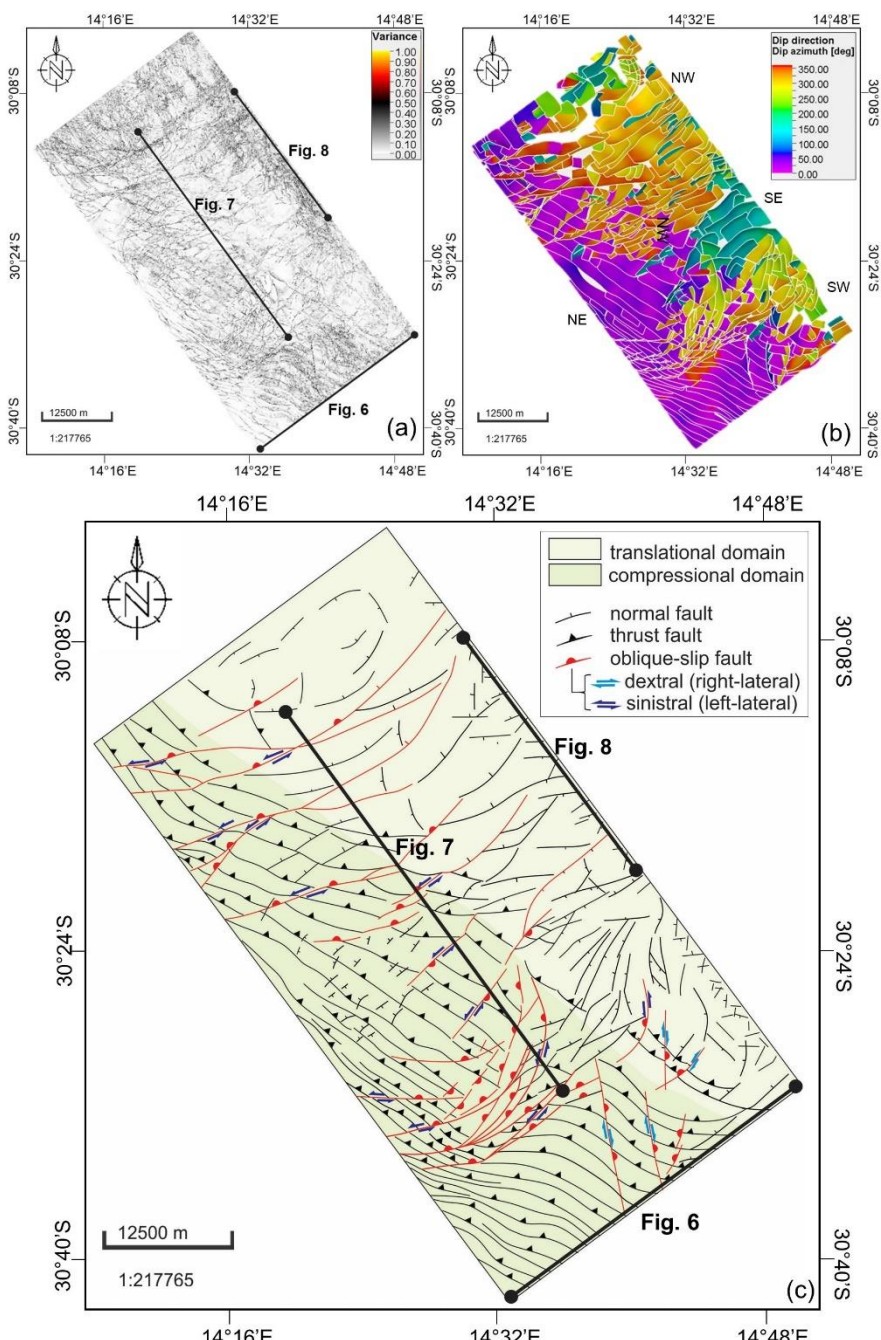


**Figure 5: Structural framework showing the translational and compressional domains imaged in the study area. (a) Uninterpreted variance time slice at -3 424 ms (imaged within sequence B2 covering depths approximately between -3 200 m to -5 000 m) showing the fault continuities. (B) the dip azimuth of various faults. (C) The interpretation of structural framework in plan view using a) and b) together with fault displacement characteristics observed in plan section.**

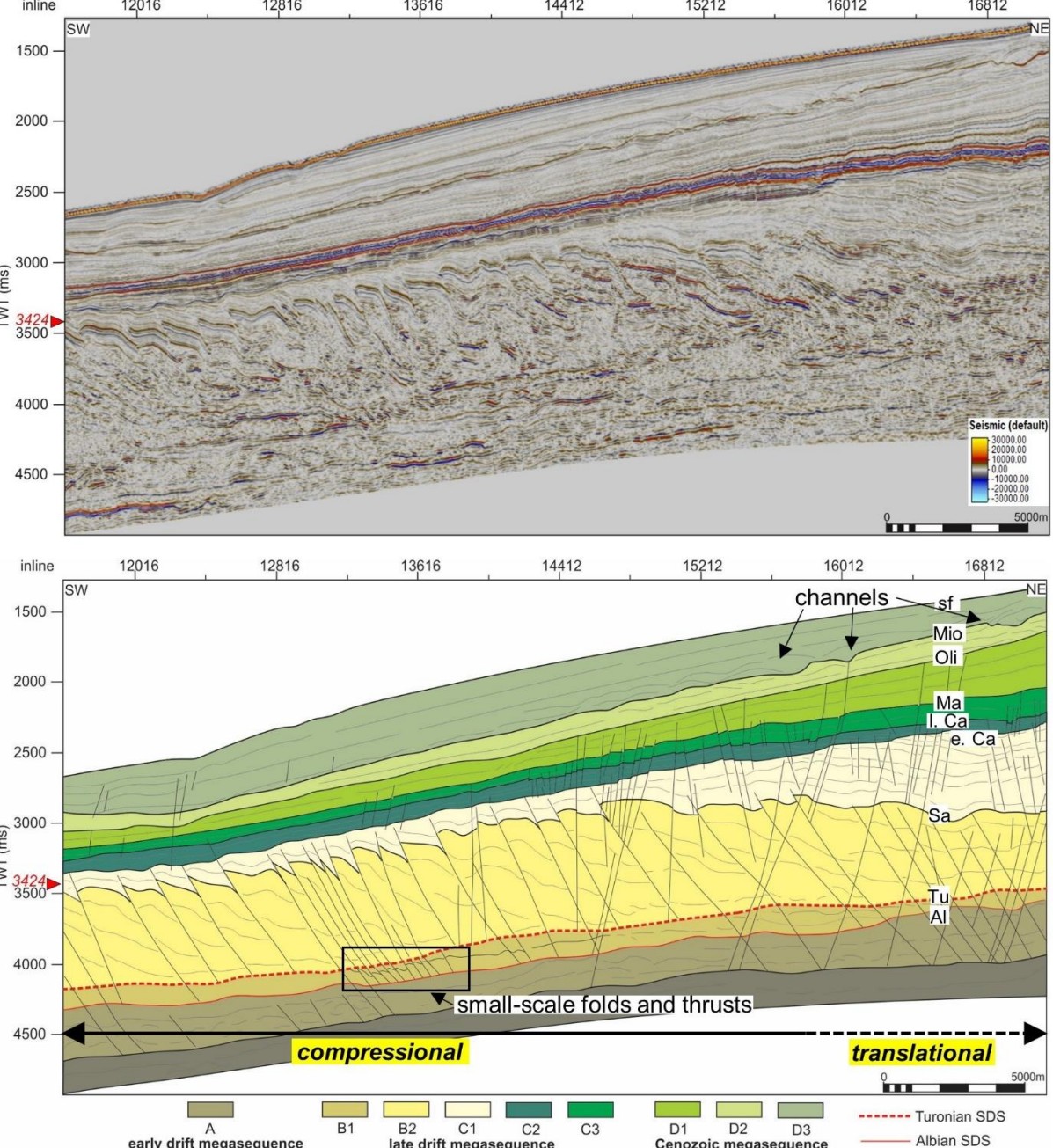

**Figure 6: Uninterpreted and interpreted sections of crossline 11544 showing a portion of the translational and compressional domains and position of timeslice 3424. The crossline lies roughly perpendicular to the compressional domain DWFTBs giving a view of their structure in comparison to the translational domain. Abbreviations: Al= Albian, Tu= Turonian, Sa= Santonian, Ma= Maastrichtian, Oli= Oligocene, Mio= Miocene, sf= seafloor, SDS= shale detachment surface. Vertical exaggeration= 5.**



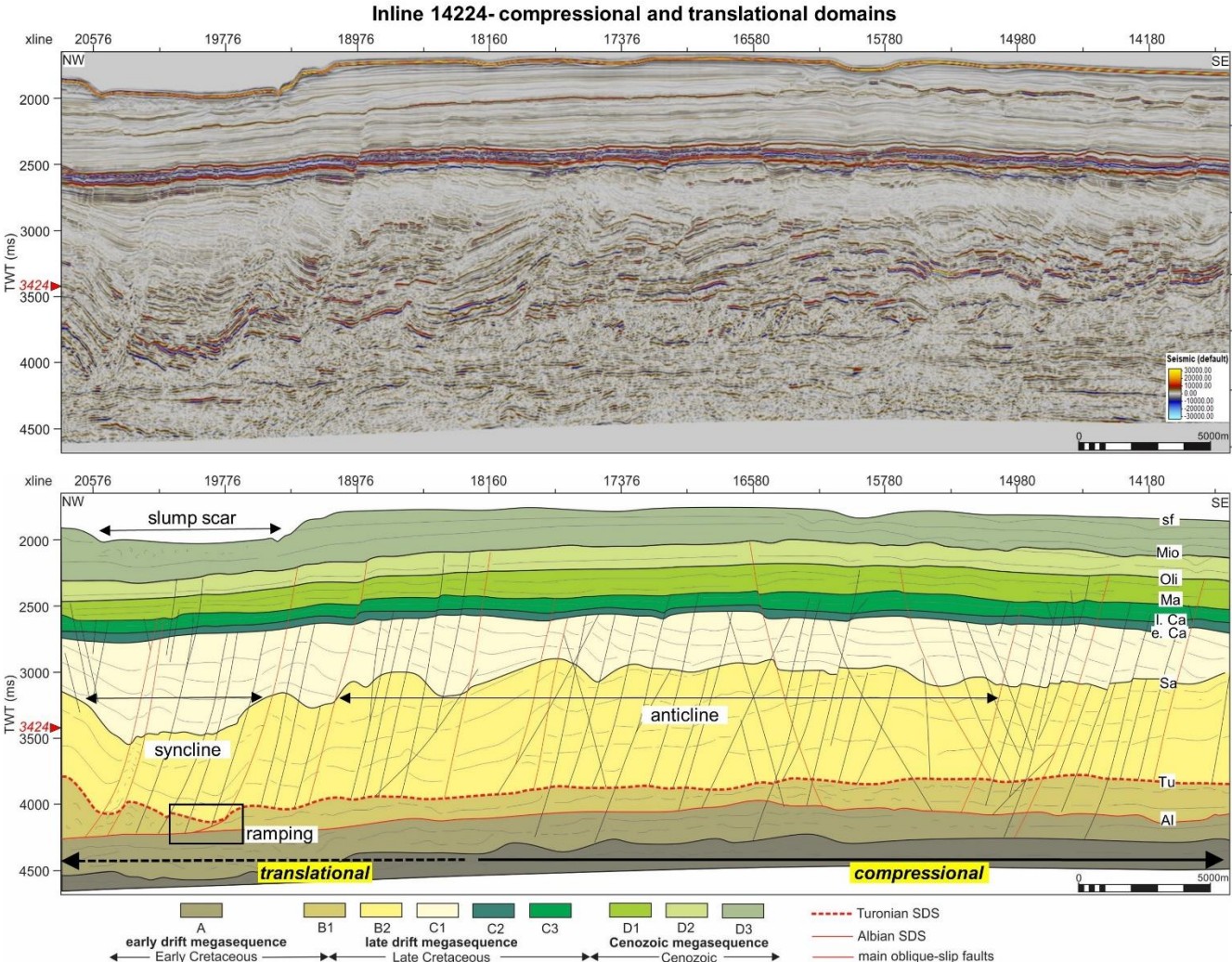

**Figure 7: Uninterpreted and interpreted sections of inline 14224 showing the compressional domain and position of timeslice 3424. The section lies roughly along-strike the DWFTBs giving a view of the oblique-slip faults that crosscut them. Present day slumping is evident above a Late Cretaceous syncline. Abbreviations: Abbreviations: Al= Albian, Tu= Turonian, Sa= Santonian, Ma= Maastrichtian, Oli= Oligocene, Mio= Miocene, sf= seafloor, SDS= shale detachment surface. Vertical exaggeration= 5.**


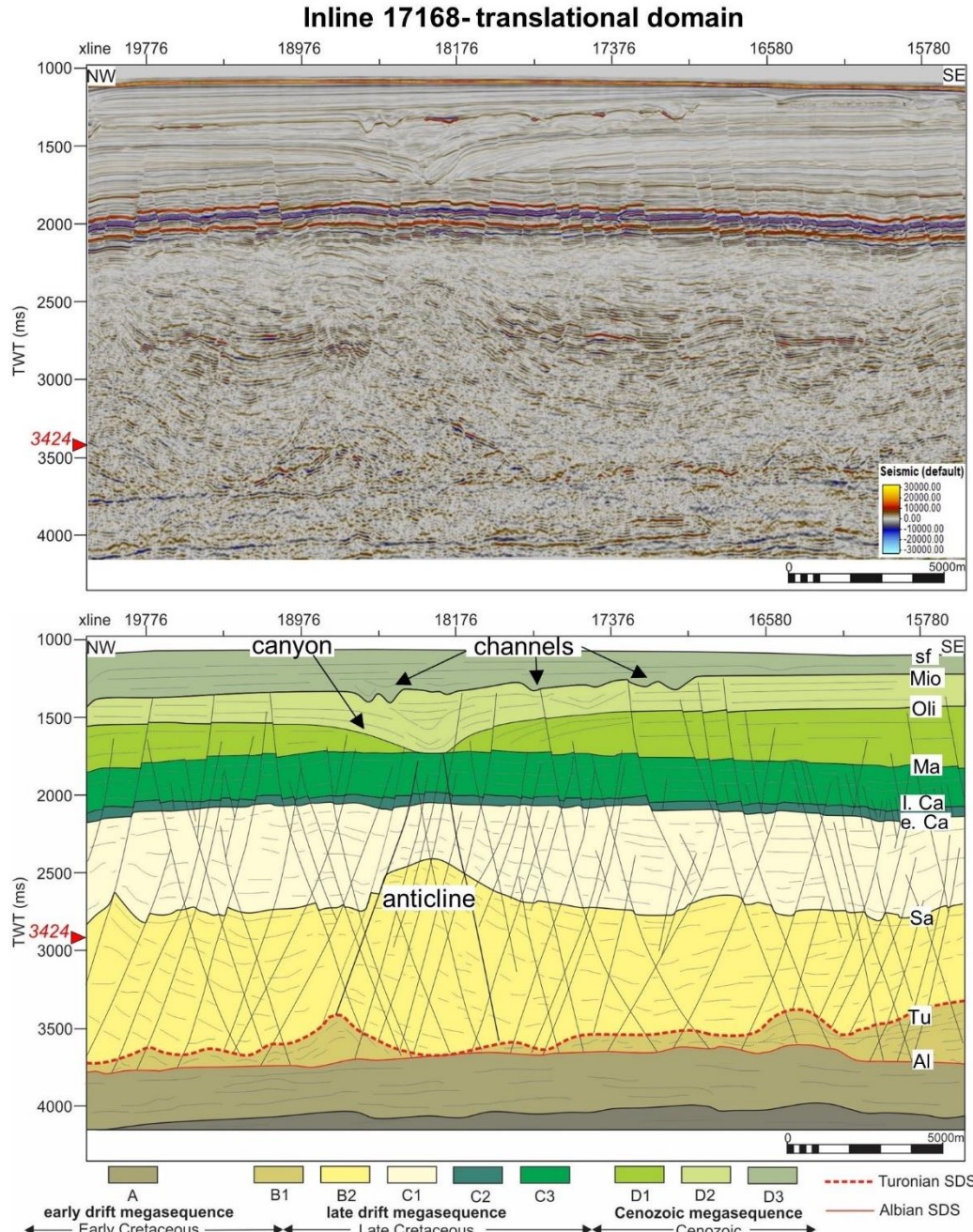

**Figure 8: Uninterpreted and interpreted sections of inline 17168 showing the translational domain and position of timeslice 3424. Normal faults extend to the Albian surface at depth and upwards into the Cenozoic megasequence with some terminating at Oligocene or Miocene unconformity surfaces. Abbreviations: Al= Albian, Tu= Turonian, Sa= Santonian, Ma= Maastrichtian, Oli= Oligocene, Mio= Miocene, sf= seafloor, SDS= shale detachment surface. Vertical exaggeration= 5.**

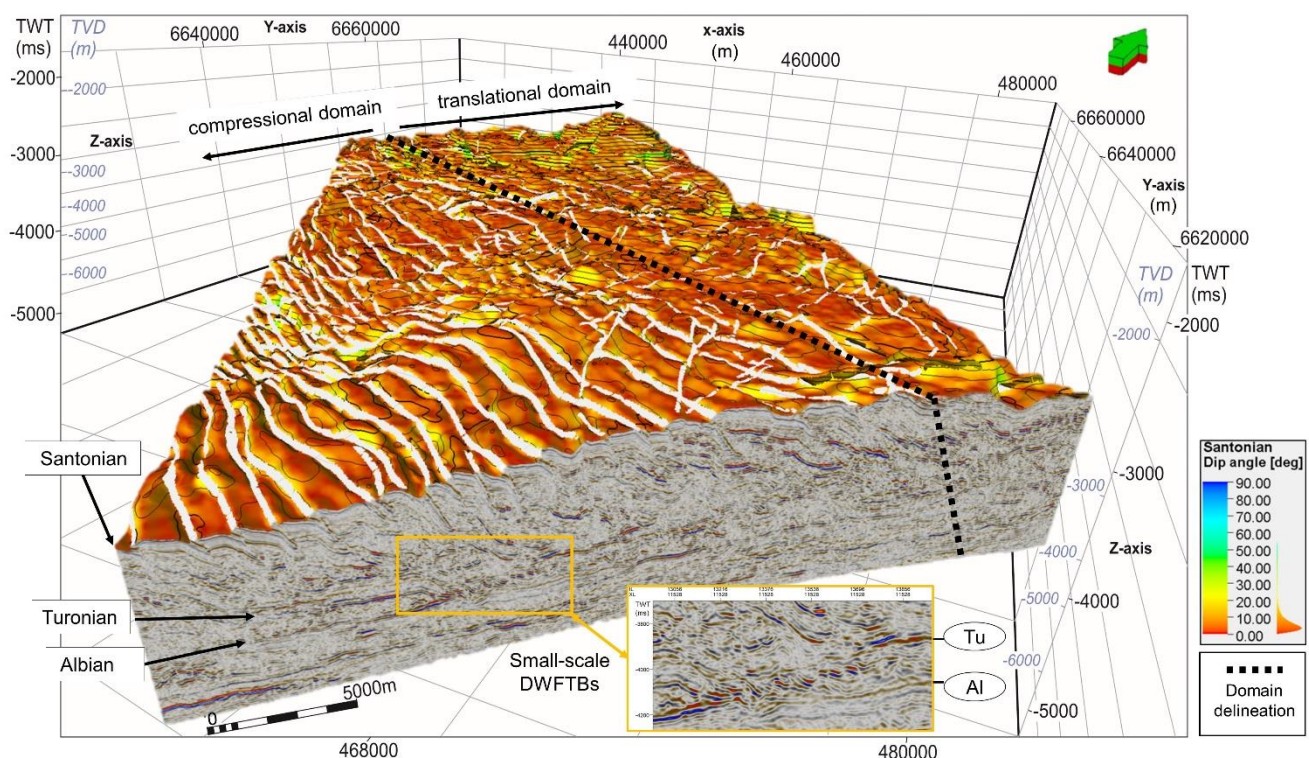

**Figure 9: 3D view of a crossline showing; an extensive, kilometre-scale DWFTB system detaching the Turonian slip surface, the Santonian horizon (in dip angle) defining the crest of the large folds, and an underlying localized set of secondary, small-scale DWFTBs upon the Albian slip surface. Vertical exaggeration= 5.**

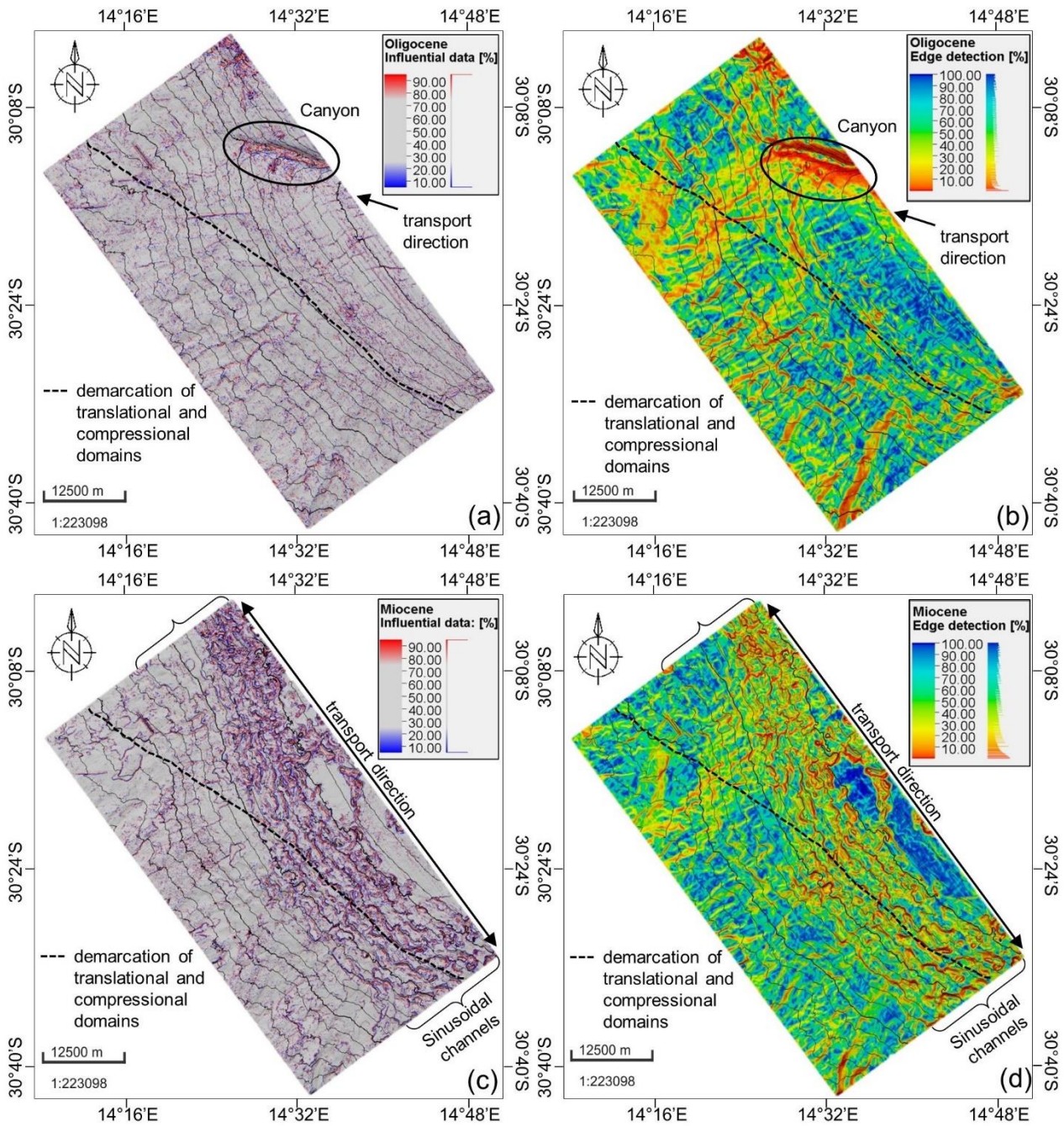


**Figure 10: Cenozoic horizons showing the canyon-channel systems. Oligocene canyon formed by a downslope turbidity current viewed using the (a) influential data and (b) edge detection attributes. Miocene sinusoidal channels formed by an along-strike bottom current viewed using the (c) influential data and (d) edge detection attributes. Vertical exaggeration= 5.**

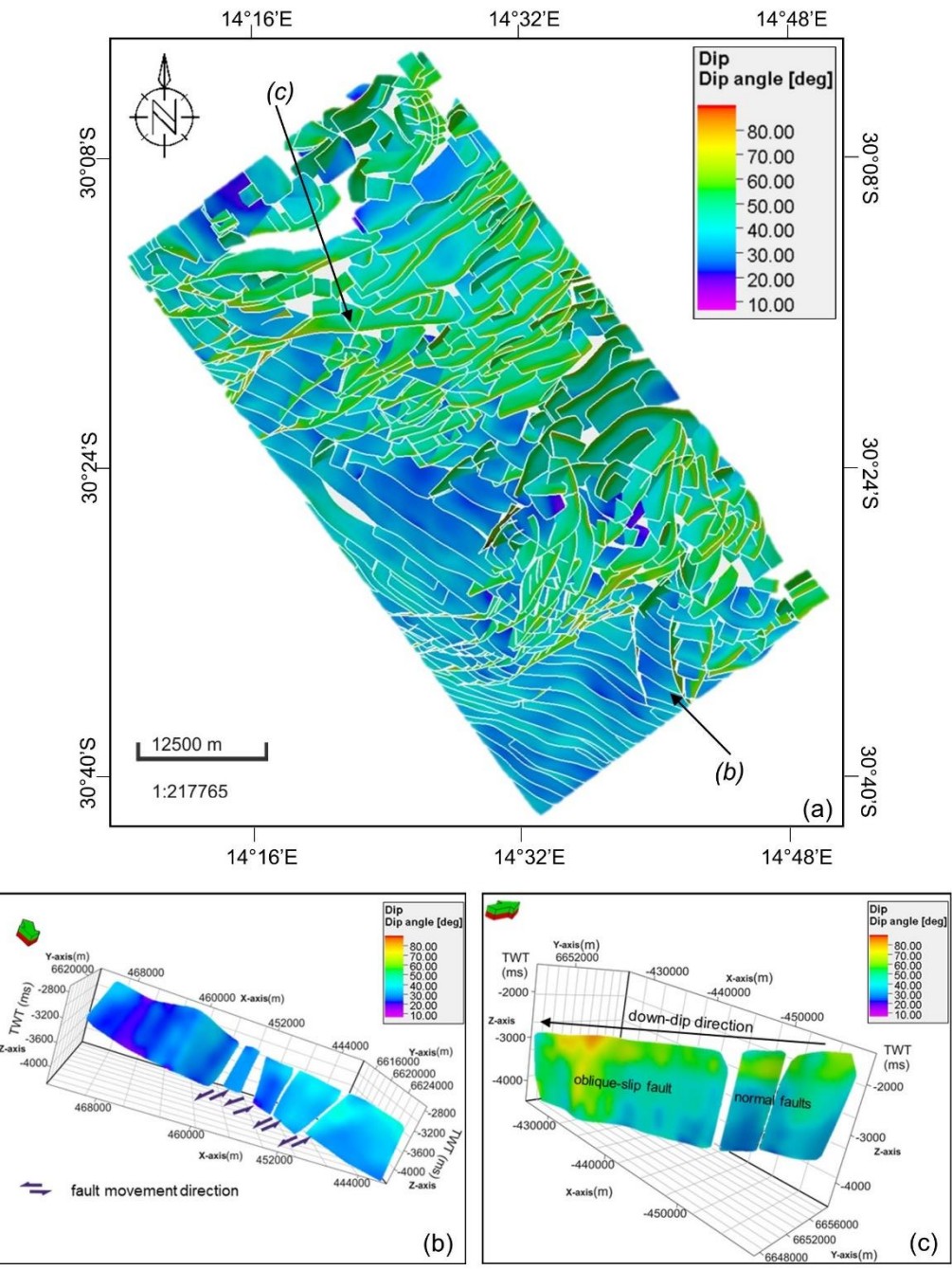


**Figure 11: Structural framework faults interpreted in the study area showing: (a) plan view of all dipping faults ; (b) segmented shallow-dipping thrust fault showing left lateral (sinistral) slip motion of displacement in 3D; and (c) steeply-dipping normal and oblique-slip fault extending between both the translational and down-dip compressional domains in 3D.**

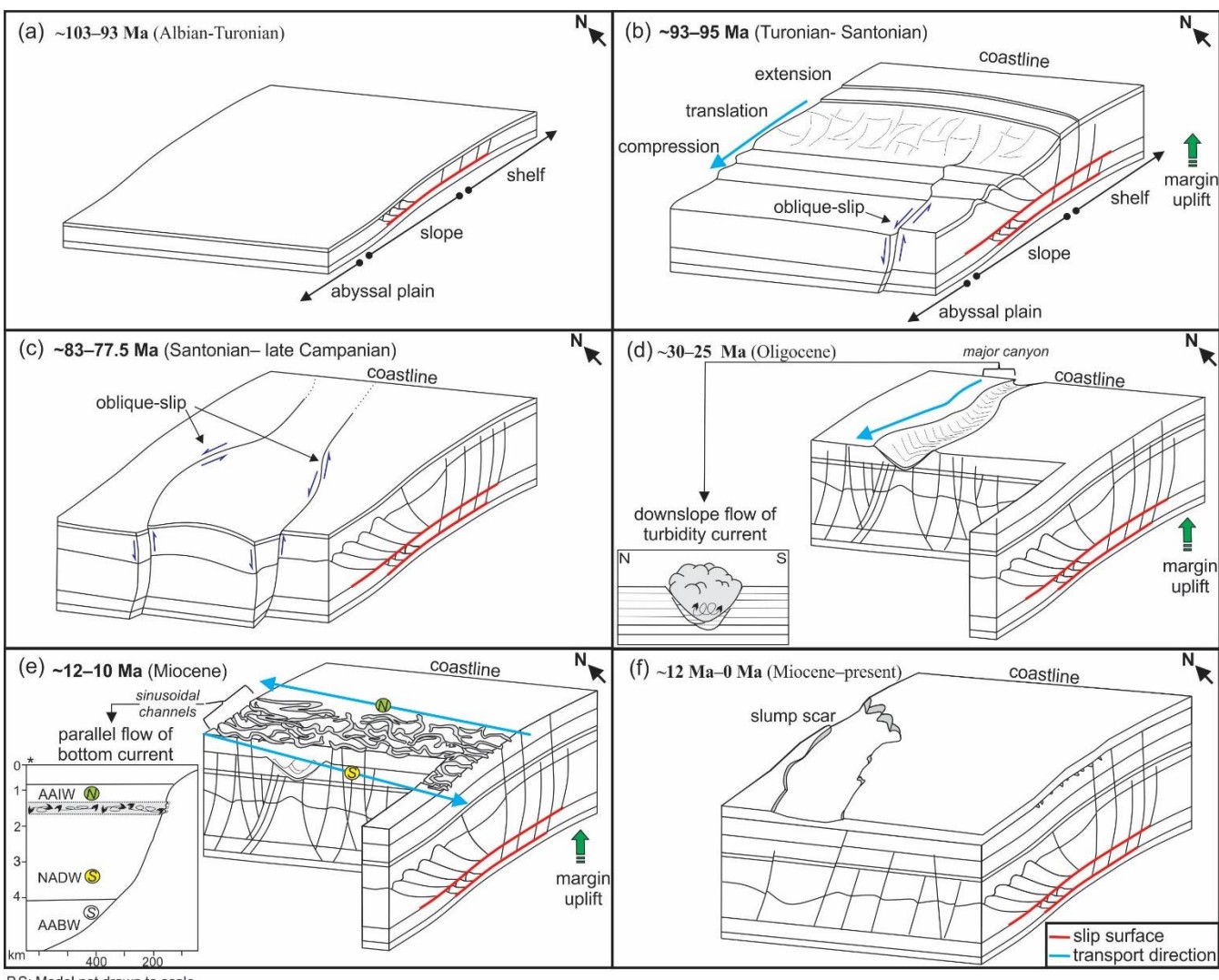

**Figure 12: Temporal model of the evolution of the deep-water Orange Basin from the Late Cretaceous DWFTB system to the mass-transport complexes of the Cenozoic. For a detailed explanation the reader is referred to Section 5.4. *Bottom current diagram modified after Weigelt and Uenzelmann-Neben, (2004).**