# Peer review of "Strato-structural evolution of the deep-water Orange Basin: Constraints from high-resolution 3D seismic data"

_EGUsphere, 2022_

## Referee Comment (RC2)

[referee-annotated manuscript omitted]

---

## Author Comment (AC3)

**Reply to reviewer comments:**

We would like to thank the editorial staff at EGUsphere for inviting us to submit a revised version of our manuscript (egusphere-2022-180). We have carefully worked through all the comments from the open discussion as closed on the 14th of August (2022) and herein provide a revised version of our manuscript. We would like to acknowledge the reviewers for their constructive comments. We compiled all comments from the reviewers, as well as our replies (in blue text) to each comment, which are presented below.

**Reviewer 1:**

Using a set of 3D seismic reflection data this manuscript aims at constraining the strato-structural evolution of the transitional and compressional domains of a Late Cretaceous deep-water fold-and-thrust belt system and its influence of the overlying Cenozoic megasequence in the Orange Basin offshore SW Africa.

The writing, wording and phrasing very often is unclear. The authors often are not precise and leave room for misunderstandings. It, e.g., remain unclear what improvement in scientific knowledge will be achieved by the study. The authors cite a large amount of literature describing the tectonic setting and development (see, e.g., Seismic stratigraphy and Results and interpretation). The interpretation of sequences and stratigraphic markers is mainly a set of statements, there is no discussion. Most of the facts presented are cited from other studies – so what is new? This is really difficult to identify.

Reply: We have fixed and restructured the manuscript.

Furthermore, the authors are very focused on the tectonic influence on sedimentation and deposition. They large ignore the effect oceanic currents and water mass transport have on this. The authors only briefly venture in this direction but in an oversimplified way and they appear to lack an understanding of physical oceanography in general and of this region in particular.

Reply: We have included the effect of oceanic currents both as a Palaeoceanography section (Section 2.4) and in the Discussion (Section 5.3).

The authors speak of high-resolution seismic reflection data. However, the data were collected with a sample rate of 2 ms (see Table 1, Nyquist frequency 250 Hz) and then resampled to 4 ms (see Table 2, Nyquist frequency 125 Hz). This certainly is not high-resolution! High-resolution seismic reflection data should be recoded at minimum 1 ms sample rate giving a Nyquist frequency of 500 Hz. And where is the use of lowering the vertical resolution of seismic data by resampling?

Reply: We received the *already processed* data from Shell and only carried out the interpretation using Schlumberger's Petrel software. Statements of it being "high-resolution" have however been removed.

The paragraph on Seismic resolution is very simplistic and, in parts, wrong. A reputable reflection seismologist uses l/2 and not l/4 (or even l/8) to compute the vertical resolution as l/4 is a very theoretical value. I would like to see a spectrum of the seismic data since I seriously doubt that the dominant frequency observed is 50 Hz; please, provide evidence for this! Even so, the Fresnel Zone definitely is not half of the dominant wavelength but dependent on target depth! Please, see Yilmaz (2001), P. 1803. There the formula is given and states

that . So, the Fresnel Zone is dependent on target depth z and the main frequency. So, even if we take the main frequency to be 48 m (serious doubts here) the Fresnel Zone in 1000 m water depth will be 155 m and in 2000 m water depth it will be 220 m. So, the assumptions of the authors of a constant Fresnel Zone of 24 m are wrong!

Reply: Thank you for the comment.

- After careful inspection of the frequency spectrum of the migrated data, we noted that the data has a dominant frequency of ~20 Hz, with a significant range of good signals between 10 - 60 Hz. Although the Orange Basin 3D Pre-processing and PreSDM 2013 report suggest that the dominant frequency on the raw data (image below) is 50 Hz, this is probably changed by a number of factors such as filtering, gain control, decon, etc. by the team that processed the data. This is migrated data so the section on the Fresnel Zone is not applicable.
  - So f = 20 Hz = 0.05 s; v = 2400 m/s; Dominant wavelength = 120
  - Vertical resolution = 30 m & 60 m for ¼ and ½ length criterion, respectively
  - Horizontal resolution = 120 m
- Nevertheless, all features described are generally much larger in scale than the re-calculated seismic vertical and horizontal resolution limit.
- Please see Figure below of a shot gather before and after the application of deghosting from the processing report:

[Figure]

Figure 8:     Spectra along a gather near(blue) mid(pink) far(orange). Left before angular deghosting right after.

- The second of these images above is what we observe in the Petrel Schlumberger software for each of the regional lines described (Fig. 6, 7 & 8 in manuscript).

[Figure]

[Figure]

The data have undergone severe processing applying several deconvolution techniques, resampling, attenuations, and time variant scaling (see Table 2). This all may have enhanced the signal but it also destroyed the true reflection amplitude. Still, the authors interpret variations in reflection amplitude both laterally and vertically (e.g., line 176, 196, 212). This, however, is no longer possible.

Reply: We received the *already processed* data from Shell and only carried out the interpretation using Schlumbergers Petrel software. Unfortunately, we disagree with the statement made in regards to the reflection amplitude. Even if the true reflection amplitude was destroyed by the processing applied on behalf of Shell, we interpret our reflection amplitudes in relation to the rest of the seismic volume - e.g. the seafloor, and other stratigraphic markers/surfaces are clearly higher in amplitude than the sequences they separate.

There is more evidence that the authors are not on top of the seismic method. They have labelled the twoway travel time negative. That implies that the data have been recoded before the shots were trigger. This is nonsense (sorry for being so blunt). It is neither the habit to label depth in negative numbers. If you want to make clear that this is below seafloor, please use mbsf (metres below seafloor).

Reply: This has been fixed and updated.

The authors use seismic attributes to aid their interpretation. They state that 'seismic attributes are designed through mathematical manipulation' but they are not designed but computed. And I hope the data have not been manipulated (even though the severe processing certainly manipulated the data).

Reply: The statement on seismic attributes has been rephrased. No, the data was not manipulated. The processing of the seismic data was not carried out by the authors in this study.

Taking all this into account I have huge doubts about the interpretation of the seismic reflection data presented by the authors since they do not appear to have fully understood the concept and physics of the method or the possibilities and limits.

Reply: This has been addressed in the previous replies. Changes have been made in section 3.2.1. Seismic Resolution Limit.

The interpretation of the data is presented before this is discussed. To me this appears awkward.

Reply: We have completely reworked the results and discussion sections. Specifically, interpretations have been separated from the Results (Section 4) and now form part of the Discussion (Section 5).

Little information is provided on the lithology. The authors just state that, e.g., a) the Albian surface represents a maximum flooding surface (how do they know this?) and b)is a shale detachment surface. This is just one example. What are the arguments for those interpretations? What knowledge is used and not presented?

Reply: Unfortunately, there is no well data in our deepwater study area according to the Petroleum Agency of South Africa (PASA). All well log data available for the offshore South African margin are shown PASA's Geoportal:

https://geoportal.petroleumagencysa.com/Storefront/Viewer/index_map.html

The knowledge used with regards to lithostratigraphy (ages and sedimentary sequence) is based on those recognized elsewhere in the Orange Basin from various previous studies. The chronostratigraphic chart shown in Fig. 4, modified from the PASA, 2017 brochure, is also used and referred to in the text. All this has been addressed in the added Section 3.2.4 (Seismic interpretation strategy).

Section 4.1.3 how does especially the younger Cenozoic tie in with other studies from this margin, e.g., (Hopkins and Cartwright, 2021; Weigelt and Uenzelmann-Neben, 2004, 2007a, b)? The discussion here is very centred on tectonically influenced sedimentation. It may be useful for the authors to broaden their reading.

Reply: This section has been completely re-worked with an additional focus on the younger Cenozoic sequences. We do appreciate this comment, however the main focus of this study was on how the geological structures in the area have influenced sedimentary sequences. We hope this addresses your concern.

Section 4.2.1 is very focused on complex attributes. In my opinion the authors got a bit carried away by the figures of the complex attributes and lost their feeling for resolution and other limits.

Reply: We have fixed and restructured the manuscript.

The whole discussion is a repetition of other people's work.

Reply: This has been fixed and updated. We would like to point out, however, that replication of results is an important (yet frequently dismissed) part of scientific research. We therefore do deem it important to validate previous work, while, of course, adding our own discoveries/interpretations.

Section 5 to me does not show any advance in scientific knowledge.

Reply: This has been fixed and updated. Also see previous comment.

It would be helpful if the described features all were marked in the figures.

Reply: This has been fixed and updated.

The References needs checking. For several references journal name, etc is missing, some references are listed twice. Jungslager, 199 is missing from References

Reply: Thanks. References have been updated.

Figure 3: Seismic depth limit is supposed to mean maximum penetration? Annotate the seismic markers discussed in the text please with the numbers used in the text

Reply: Figures have been updated with stratigraphic markers following those recognised for the Orange Basin (Brown et al., 1994). The seismic markers have been annotated as they appear in text (as their ages - e.g., Maastrichtian, Campanian, Santonian, etc.) - rather than numbers to avoid confusion. Thank you for your comment.

Figure 4 is obsolete

Reply: It has been removed and the interpretation methodology is briefly explained in text- Section 3.2.

Figure 5: How were the type of the faults and the throw direction identified? Certainly not based on Fig 5a, which is rather chaotic. Travel time can never be negative! Show coordinates

Reply: The fault throws and directions were identified from previously Figs. 12a and b (plan section of all faults). The type of faults were identified through interpreting the seismic data in 3D- e.g. hanging wall vs footwall vs strike-slip displacements were taken into consideration. To make this clearer we reorganized the images.

Figs. 6-8 TWT never is negative! Same applies to depth; if needed use mbsf

Reply: Fixed. Estimated depths have been removed.

Figs 9-12 are extremely confusing and not helpful at all. Please, omit

Reply: We are not in total agreement. To remove confusion the sequence of figures in the text was changed:

- **Fig. 9** is now omitted since all surfaces are shown in the 2D seismic sections already (Figs. 6-8).
- **Fig. 10** remains. Shows the crest of thrust folds and the NW-SE trending thrust faults well.
- **Fig. 11 and b** is the plan view of Miocene and Oligocene horizons which show major erosional features; sinuisoidal channels and a large submarine canyon respectively. Omitting this would also mean that we need to omit it from the results and discussion. The 2D seismic sections showing these two horizons (Figs. 6-8) do not show the full extent of erosion.
- The faults in plan view from **Fig. 12 a and b** is how Fig. 5 was created. To avoid confusion the figures have been rearranged in better sequence.

Figure 13: How were the ages assigned? Where does the anticline in 13d originate? Cold water usually has a higher density than warm water (annotation 13e)

Reply:

- The knowledge used with regards to lithostratigraphy (ages and sedimentary sequence) is based on those recognized elsewhere in the Orange Basin from various previous studies. The chronostratigraphic chart shown in Fig. 4 modified from the PASA, 2017 brochure is also used and referred to in the text. All this has been addressed in the added Section 3.2.4 (Seismic interpretation strategy). Ages are estimates of those assigned to each surface as deduced from literature. Literature include Séranne and

Anka, 2005; Paton et al., 2007; Wigley and Compton, 2006; De Vera et al., 2010; Hirsch et al., 2010; Kuhlmann et al., 2010; Scarselli et al., 2016, etc.

- The anticline is shown in Fig. 8.
- Fixed the geological model.

Line 40 The authors state that this margin is largely underexplored since there is only one well per 4000 km². This is meant regarding hydrocarbon exploration? Extensive sets of seismic reflection data have been collected along the margin and several scientific sites have been drilled. So, regarding the development of the passive continental margin a wealth of information is available.

Reply: Yes, in South Africa the deepwater Orange Basin is underexplored in contrast to Namibia and the shallow shelf environments. The authors added this for clarity in the Introduction (Section 1).

Lines 44/45 what is the scientific importance of this?

Reply: We have fixed and restructured the section.

Line 46 what is 'early' 2D seismic data

Reply: This has been fixed and updated.

Lines 48/49 already said in line 46

Reply: This has been fixed and updated.

Lines 58/59 'an in-depth examination of the transitional domain from a buried DWFTB system' – from the DWFTB to what? Confusing

Reply: We have fixed and restructured the section.

Line 93 this is a very confusing sentence. Rephrase

Reply: Rephrased.

Line 102 this is only true until the onset of the Benguela Current in the Miocene. Please, study the right literature

Reply: Thank you for your comment. We have added a subsection called 'Palaeoceanography', under the regional setting section, which details oceanic currents in the area of the Orange Basin.

Line 117/118 why was only a subset of the 3D survey used for this study? No argument given

Reply: It's simply too much data (~60 km × 120 km) for our current computing equipment. Specifically, the Petrel software and computer equipment struggled with the size of the dataset. Additionally, processing and interpreting the whole dataset was not necessary for this study as the main structural framework and features of interest are already clearly shown.

Line 125 why were the data resampled to 4 ms? That reduced the vertical resolution by half!

Reply: We did not carry out the seismic processing. Seismic processing was done by the Netherlands Global Processing Team on behalf of Shell.

Line 129 the velocity model used for depth conversion is rather crude. How was this derived? What is the reason for not using velocities derived during velocity analysis?

Reply: That is true. The velocity model, and hence, depth conversion was crude at best. The velocities were derived from Kuhlmann et al. (2010) from well logs in the shallow Orange Basin. We chose to omit the rough depth conversions from Figs. 6-8 to avoid unnecessary errors.

Line 151 'local deviation of the seismic signal' – from what?

Reply: local deviation *from* the signal.

Line 170 'often always' – which one?

Reply: Often. We removed the 'always'.

Line 248ff how was constraining the ages carried out? And here it is stated that age assignment was difficult, but in the previous paragraphs ages were used very confidently…. It really would be nice to see a correlation of seismic data with ages and lithology at a well.

Reply: The knowledge used with regards to lithostratigraphy (ages and sedimentary sequence) is based on those recognized elsewhere in the Orange Basin from various previous studies. The chronostratigraphic chart shown in Fig. 4 modified from the PASA, 2017 brochure is also used and referred to in the text. All this has been addressed in the added Section 3.2.4 (Seismic interpretation strategy). Ages are pure estimates of those assigned to each surface as deduced from literature. Literature include Séranne and Anka, 2005; Paton et al., 2007; Wigley and Compton, 2006; De Vera et al., 2010; Hirsch et al., 2010; Kuhlmann et al., 2010; Scarselli et al., 2016, etc.

Line 268 how do you know the Oligocene unconformity was formed subaerially?

Reply: This has been amended in the text and updated.

Line 270 or other mass transport?

Reply: Yes. Or other type of mass flow. This has been amended in the text and updated.

Lines 278-280 this cannot be seen in the figure

Reply: This is unfortunate. We believe this can be seen well in Figs. 6 and 8. The basal deposits above the Miocene stratigraphic marker are initially mounded.

Lines 280-282 see, e.g., (Weigelt and Uenzelmann-Neben, 2004)

Reply: Thank you for the reference. Slumping in our study is seen to affect up to the seafloor and not just ~early Pliocene.

Line 285 TWT can never be negative!

Reply: Fixed.

Lines 289-290 it is unclear how this detailed interpretation was derived

Reply: We fixed this accordingly. Fig. 5 was constructed using the dip azimuth of all faults, the variance time slice and viewing the actual data in 3D. We hope the updated Fig. 5 is enough to show how the interpretation was derived.

Line 292 'as explained previously' – this was not explained but stated

Reply: This has been fixed. The manuscript has been restructured.

Lines 294-296 and what is the importance of this?

Reply: This is discussed in section 5. The importance is that it shows fault renewal by margin instability.

Line 305 it is impossible to see this in the somewhat chaotic Figs 9 and 12

Reply: Figure 9 has been removed from the manuscript. The interpretation and rephrasing has been fixed.

Line 310 to me the spoon-shaped feature looks over-interpreted

Reply: Noted and fixed - we removed the delineation. We described the observed structural framework as being oval-shaped purely only on plan view. Otherwise, we just referred to the "central" zone when talking of this middle region to guide the reader.

Line 314 in which sequence?

Reply: In sequence B2, Fig. 7.

Line 321' deviation from the normal trend' – what is the normal trend?

Reply: The "normal" trend was left-lateral sinistral slip motion. We agree this sentence was poorly worded.

Line 326 the authors cannot resolved metre-scaled displacements! The resolution of the seismic data does not allow this.

Reply: That is indeed true. This was fixed to simply "displacements below the seismic resolution limit".

Line 329 mounded and chaotic geometry is not necessarily a sign for turbidites. There are other mass transport deposits

Reply: That is indeed true, however, it is the most plausible for the Albian to Turonian intervals which is what we interpreted.

lines 335-337 this is no discussion!

Reply: We have restructured the manuscript and these lines were added into what we feel is now a proper discussion.

Lines 341-343 repetition from lines 335/336

Reply: Noted and fixed. This manuscript has been restructured. Some of these points now belong in the discussion.

Lines 351-353  what does this mean? Confusing

Reply: We rephrased this to hopefully avoid confusion.

Lines 372-394 this is all other people's work but the manuscript rests on this. How were the ages identified?

Reply: The manuscript has been restructured and our geological model (Fig. 12) has been updated based on our observations including that of previous literature. The knowledge used with regards to lithostratigraphy (ages and sedimentary sequence) is based on those recognized elsewhere in the Orange Basin, from previous studies. The chronostratigraphic chart shown in Fig. 4 modified from the PASA, 2017 brochure is also used and referred to in the text. All this has been addressed in the added Section 3.2.4 (Seismic interpretation strategy).

Lines 417-420 there is no anticline in Fig. 7

Reply: Since synclines form together with anticlines, there is an anticline; the rest of sequence B2 sediments in Fig. 7 is a broad anticline. We therefore add this interpretation to the figure to make it clear.

Lines 439-442 all this is based on already published studies

Reply: The replication of results is an important part of scientific research to validate previous studies and adding our own discoveries/interpretations.

Line 454 the canyon in Fig. 8 is definitely not fault controlled

Reply: That is true and has been fixed.

Line 463 there are plenty of more recent studies based on higher quality data than Dingle et al., 1983

Reply: Dingle et al., 1983 has been removed.

Line 465ff this whole discussion is focused on tectonic sealevel variations. How does the onset of Antarctic glaciation tie in, how the variability in Antarctic ice-sheet thickness and size, the variability in the location of the Southern Ocean frontal systems?

Reply: We have added this into the discussion. Thank you for your comment.

Line473 which existing planes of weakness? What caused them?

Reply: This has been removed since the canyon is not fault-controlled. Rather, faults were hindered by the canyon.

Line 484 it is interesting to see that the authors cite a paper from the Brazilian margin. There, the oceanographic system is quite different to the SW African. Why not cite papers which dealt with the SW African margin?

Reply: The reference on the Brazilian margin has been removed from our discussion.

Line 480ff it appears to be assumed that deposition is mostly fault controlled. That is not the case for SW Africa, where upwelling, NADW, AABW and the Benguela Current significantly influence sediment transport and deposition.

Reply: Thank you for the contribution. This has been fixed in the interpretation of the Cenozoic.

Lines 490ff the ocean offshore SW Africa is strongly stratified, which results in baroclinic/geostrophic flow. Internal waves are not needed for this. Here, the authors think too

complicated. I seriously doubt that tidal movements will affect deposition/erosion in 2000 m water depth. Tidal current further act not slope parallel.

Reply: Thank you for your comment. This has been fixed in the interpretation of the Cenozoic.

Line 492 what is erosional undercutting?

Reply: This has been removed and the sentence rephrased.

Line 494 differences in temperature/salinity generally cause geostrophic flow, not only parallel to the slope

Reply: Noted. The text has been fixed and updated.

Lines 497-501 those erosional features may have been formed by AABW or NADW, not only by upwelling. Also see (Weigelt and Uenzelmann-Neben, 2004)

Reply: Thank you for the reference. It did indeed help guide our interpretation of the Miocene slope-parallel channels.

Line 501 and due to variability in the glaciation of Antarctica!

Reply: This has been added.

Line 502 in what water depth?

Reply: We feel this is stated in the following sentence. According to Compton and Wiltshire (2009): "*bottom current data from current meters placed at intervals along the 1000 m contour from Lüderitz*" and "*A current meter placed at 3000 m water depth in the Cape Basin*"...

Lines 503-505 this is NADW!

Reply: Noted and added, thanks.

Line 509 a CDS show high sedimentation rates!

Reply: This interpretation has been reassessed.

Line 512 it is not true that erosional features of CDSs have been poorly studied

Reply: We have removed the interpretation of the CDS.

Lines 513-514 e.g., (Weigelt and Uenzelmann-Neben, 2004)

Reply: This reference and others referred to have been added to the discussion.

Lines 549-552 I do not agree. Most of the following was already known previoulsy

Reply: Since we have restructured our manuscript and added our own discoveries/interpretation, we believe our research *does* shed light onto the kinematics, geometry and displacement characteristics of DWFTB systems and furthermore adds to the literature on ocean currents affecting Cenozoic sedimentation. We would like to point out that it is also scientifically important to validate (i.e., replicate) previous work.

Hopkins, A., Cartwright, J., 2021. Large scale excavation of outer shelf sediments by bottom currents during the Late Miocene in the SE Atlantic. Geo-Marine Letters 41, 33.

Weigelt, E., Uenzelmann-Neben, G., 2004. Sediment deposits in the Cape Basin: Indications for shifting ocean currents? AAPG Bulletin 88, 765-780.

Weigelt, E., Uenzelmann-Neben, G., 2007a. Early Pliocene change of deposition style in the Cape Basin, southeastern Atlantic. Geological Society of America Bulletins 119, 1004-1013.

Weigelt, E., Uenzelmann-Neben, G., 2007b. Orbital forced cyclycity of reflector strength in the seismic records of the Cape Basin. Geophysical Research Letters 34.

Reply: Thank you for the suggested literature. It indeed helped in understanding the role of ocean currents and upwelling accounting for the mid Cenozic to present day stratigraphic sequences.

**Reviewer 2:**

The authors of this manuscript are to be congratulated for presenting some very elegant interpretation and visualisations of an interesting data set that images a the compressional and transitional (or translational?) domains of a large scale submarine slide complex.

I think that there are a few issues that the authors should consider in their interpretation and discussion of their observations. I feel that some of the inferences are somewhat circumstantial, and it would be useful if they could be better substantiated, or the alternatives considered. They are:

- Evidence that the small-scale underlying thrusts in the Albain sequences are younger than overlying large scale thrust system. This seems counterintuitive, particularly given that thrusts normally cut up stratigraphy, particularly when a basal detachment becomes "locked". Why could these older thrusts not simply be part of an older mass transport complex, which would be an equally valid (and more probable) interpretation of the data and the obsereved relationships?

Reply: Interpretations have been modified in the text and Figure 12 of the model.

- The use of the term transitional rather than translational to describe the domain between the extensional and compressional domains of a mass transport complex.Both are used in the literature, particularly for mass transport complexes in this region. "Transitional" may be valid where the zone is narrow, and extensional and compressional structures interact. "Translational" is more appropriate where the zone is wide and the sequences above the detachment are being displaced horizontally between the extensional and compressional domains. It think the situation described here is more akin to the latter, in which case the observations are particularly interesting. All the faults in this domain (even those interpreted as extensional) are highly oblique to the thrust faults in the compressional domain, and I suspect are also oblique to the extensional faults in the extensional domain, although this is not imaged in this data set. Are there any more extensive 2D surveys or existing maps that could be used to address this? If so, the inference that the translational domain consists mainly of oblique faults would be very interesting and innovative and would indicate the style of deformation that operates in this zone.

Reply:

- Thank you very much for the information with regards to transitional vs translational domains. We changed 'transitional' to 'translational' throughout the manuscript.

- Yes, all transitional domain faults are oblique to the thrust faults. There are no 2D seismic surveys in this study. Other studies of DWFTB in the Orange Basin may however be used to address the structural framework in the extensional domain from extensive 2D surveys that image a full DWFTB system. The most used 2D section and interpretation when referring to DWFTBs in the Orange Basin is that de Vera et al. (2010).

  · The use of the term "spoon-shaped" to describe the plan view pattern of the oblique faults – I think this is confusing, and the inference of this geometry by extrapolation beyond the extend of the data set is geologically and mechanically unrealistic. I think it is better to confine the use of the term to the three dimensional geometry of individual fault planes that show curvature in three dimensions (up dip and along strike). This has previously been used to describe the geometry of linked extensional and oblique faults in the extensional domain.

Reply: That has been fixed. The oblique-slip faults are simply concave upward. The majority dip NE, and in the compressional domain display an oval-shaped pattern/geometry in plan section.

  · The inference of the sense of motion on oblique faults from the offset of thrusts.The implication in the manuscript is that the thrusts were originally contiguous structures that have subsequently been offset by the oblique faults. I think this is unlikely. I think it is more likely that they act like lateral ramps in thrust sheets and accommodate differences in displacement between originally offset thrusts. The actual displacement will depend on the nature of the offset, and will be variable. Transform faults between offset segments of a mid ocean ridge are also a good analogy in this respect.

Reply: Thank you for your comment. This inference has been fixed in the manuscript and updated in the model - Fig. 12.

  · The use of the term "mass transport complex" to describe the turbidite and contourite deposits in the Oligocene and Miocene sequences – this is very confusing!The term mass transport complex should be restricted top large bodies of intact or semi-intact sediment transported down slope by gravitational processes, and be distinguished from sediment being transported by currents, that still may be gravitationally controlled (mass flow would be a better term for these if you prefer to avoid using terms such as turbidite and contourite). The Deep Water Fold and Thrust Belt is part of a mass transport complex, and the term should be restricted to that feature.

Reply: Thank you for your comment. This has been fixed in the manuscript. Rather, we refer to mass flow processes.

Evidence for the control of the underlying structure on the younger canyons and contourite channels.The evidence seems to suggest that the Oligocene canyon has a different orientation to the underlying structure, so it is difficult to see the control. This therefore also reduces the likely control of the underlying structure on the Miocene margin-parallel channels. It would be better to use maps that superimpose the sedimentary features on underlying structure to establish these relationships, rather than inferring them from vertical sections where apparent relationships my just be an artefact of the location of a single section. The suggestion that the margin-parallel channels are influenced by strong tides also seems somewhat circumstantial. Are there any observations from the data that can be used to support this?

Reply: This has been fixed in the manuscript, rather, we refer to mass flows in the Cenozoic. The role of parallel flowing oceanic currents plays a larger role than the underlying Cretaceous

structure as we re-interpreted the data. The margin-parallel channels we interpret are formed from the interaction of the Antarctic Intermediate and North Atlantic Deep Water currents.

Chris Elders

Curtin University

**Reviewer 3:**

Nombuso Maduna and co-authors have conducted a thorough study of a deep water fold-and-thrust belt in the Mesozoic Orange Basin, and the structural processes and features that are associated with this. The 3D seismic dataset interpreted here allows for more detailed analysis of this stratigraphy and interpretations that add to the current state of knowledge.

In this review, I recommend a careful check of how the results are presented, as sometimes they mix discussion points in this section, and that makes it difficult for a reader to discern what is new from what was previously published. I realise that this deep water fold-and-thrust belt has been published on before, but in the introduction I felt that a short description of what it is would be helpful for clarity. The reason for this is that although this is seemingly driven by gravitational activity, at first it appears strange in a context of Gondwana break-up to read about compression. In this regard, the composition of shale is important for the structural model of detachment. Have the shales been cored or sampled? I.e., how do we know that those reflectors and units are shale? A lot of your structural interpretation is based on the properties of this rock type, so please clarify this at the beginning. In the abstract too, I recommend up front adding a statement that this refers to gravity-driven compression on an extensional margin. Generally, as well, the angle of the slope required to generate these compressional faults seems to not be too steep. Perhaps look a little into this as well.

Reply: Thank you for your kind and constructive comments. As pointed out by reviewer #1, we have completely revised the Results and Discussion sections to avoid the mixing of discussion points in the results section. We have also re-worked the introduction for clarity. As for the composition of the shale - this is unknown but inferred from well log data and literature in other areas of the Orange Basin. There are no known boreholes that have been drilled in the area. As previously mentioned, the lithostratigraphic sequence was obtained from previous studies. We agree with your comment that we should stress this and we now do so in the methods section.

Specific comments:

Line 67: replace Lower with Early. And check the paper for consistency in this regard. The difference is that referring to time only, you'd say Early Cretaceous. But describing deposits, it would be Lower Cretaceous rocks (for example).

Reply: Fixed. Thank you for your comment.

Offshore structural framework: there is a 2020 publication by Baby et al. too, that may be worth checking out: Baby, G., Guillocheau, F., Boulogne, C., Robin, C. and Dall'Asta, M., 2018. Uplift history of a transform continental margin revealed by the stratigraphic record: The case of the Agulhas transform margin along the Southern African Plateau. Tectonophysics 731, 104-130.

Reply: Thank you. We have added Baby et al., 2020 (Baby, G., Guillocheau, F., Braun, J., Robin, C. and Dall'Asta, M., 2020. Solid sedimentation rates history of the Southern African continental margins: Implications for the uplift history of the South African Plateau. *Terra*

*Nova, 32*(1), pp.53-65) and Baby et al., 2018 (Baby, G., Guillocheau, F., Morin, J., Ressouche, J., Robin, C., Broucke, O. and Dall'Asta, M., 2018. Post-rift stratigraphic evolution of the Atlantic margin of Namibia and South Africa: Implications for the vertical movements of the margin and the uplift history of the South African Plateau. *Marine and Petroleum Geology, 97*, pp.169-191) as references to the manuscript. The particular reference you referred to relates more to the southern transform margin rather than the Namibian and South African margins.

Lines 93 and 94: comprises (not comprises of)

Reply: This has been fixed and updated.

Line 101: was eustatic sea-level change. Here I would also recommend leaving out the word 'eustatic', as even into drift there was appreciable uplift and subsidence in this area.

Reply: This has been fixed and updated.

Line 115: between 2012 and 2013

Reply: This has been fixed and updated.

Line 127: data is plural of datum

Reply: This has been fixed and updated.

Methods: please clarify which were the methods you applied, and which were done by the petroleum company.

Reply: This has been fixed and updated.

Results and interpretation: I think in this manuscript it will be clearer to separate the results from the interpretation.

Reply: This comment has already been discussed above. Thank you.

Line 161: The study area lies offshore of northwest South Africa, along the continental slope

Reply: This has been fixed and updated.

Line 204: this is an example of where I am unsure whether this is your interpretation, or one from elsewhere, and why I suggest you separate the results and lay out only new ideas in that section.

Reply: This comment has already been discussed above. Thank you.

Line 214: what about synclines? Surely with anticlines there are also associated synclines. And are these anti- and synclines, or anti- and synforms?

Reply: The observed anticlines are antiformal and synclines are synformal; these distinctions have now been included in the interpretation and discussion on the stratigraphy(Section 5.1). The authors do find it awkward, however, to talk about associated synclines in the "fold-and-thrust" belt systems.

Line 219: A and B2

Reply: This has been fixed and updated.

The paragraph including line 230 is all discussion.

Reply: This has been fixed and updated.

Line 238: how did you interpret that this is a MFS? I suggest in the results, explaining the reasoning behind assigning these surfaces and units. Was it based on geometry, or truncation of what is below, for example? You could also consider tabulating this sort of information, but either way I think it is important to say something about how you arrived at your assigned surfaces. Also for the methods, which sequence stratigraphic terminology and methods did you follow and why?

Reply: We fixed this accordingly and tabulated the types of surfaces in a new Table 3. We used the sequence stratigraphic terminology proposed by Mitchum et al. (1997) for internal reflection patterns and Catuneanu (2006) for deepwater marine surfaces.

Line 279: Cenozoic unit

Reply: This has been fixed and updated.

Line 278: rather than earliest, perhaps say basal sediments within the unit

Reply: This has been fixed and updated.

In the structural framework you describe features at depths measured in both ms and metres. If you have done a time/depth conversion, please also include these depths on the figures of profiles.

Reply: The estimated depths were removed from all figures since the velocity model and hence depth conversion were crude at best. The velocities were derived from Kuhlmann et al. (2010) from well logs in the shallow Orange Basin, however, no well logs have been drilled in the present deepwater study.

Line 311: what is spoon shaped geometry? I am not sure this is a recognised term? And it is not obvious to me, without looking at the image to go with this, what that means anyway. Maybe refer to convex or concave instead.

Reply: Fixed accordingly throughout the text.

Line 315: replace 'compared to' with 'as'

Reply: This has been fixed and updated.

Lines 319 and 320: another example of mixed interpretation in the results

Reply: This has been fixed and updated.

Line 332: extremely is a bit too emotive

Reply: Rephrased.

Line 371 onward: it is not clear to me which of these findings are new (from your work) and which were established previously. I suggest laying out what the accepted model was prior to your work, and then onward from that presenting the new model based on your data and interpretation thereof.

Reply: We have completely revised the structure in the discussion to be more cohesive outlining our interpretations, then comparing/ combining what we observed with previous studies.

Line 393: this is the first time the Benguela Current is discussed and it needs to be introduced earlier in the regional setting. There is an appreciable amount of literature, in particular by Uenzelmann-Neben and colleagues, on the role of oceanographic circulation on seafloor sediments and I think your paper will benefit by including this in your interpretations and your context. This erosion is an important part of the story of deposition and preservation on this margin.

Reply: Fixed accordingly. Added a section on Palaeoceanography earlier in Section 2.4.

Line 403: differs against what? Are you referring to within the sequences, or between them, for example? Please be more clear here.

Reply: This has been fixed and updated.

Line 427: we propose a third model that

Reply: Fixed.

Line 432: 'much greater than 10 km' does not say too much. Have you got a sense of at least how long this may be?

Reply: That is difficult to estimate since it lies beyond the seismic dataset. However we do note that we see ~ 20 km of the translational domain.

Ahead of section 5.3, and following the paragraph where you propose a third scenario, I feel that a section explaining how you can get extension and compression at the same time is necessary.

Reply: We believe this all relates to the distribution of internal strain during deformation and have attempted to explain it.

Line 466: shelf, rather than coastline?

Reply: Fixed accordingly.

Line 476: which river is this canyon associated with?

Reply: Both the Orange and the Olifants rivers. This is difficult to determine since we would expect a more easterly direction of flow if it were the Orange River when comparing its relative position to the coastline and the trend of the rivers. From the Olifants River's similar NW trend it appears that the canyon was predominantly fed by it. It is important to note however that the study area lies very far from the coastline and it is unlikely that the canyon extends all the way to the mouth of the rivers as a channel formed as a subaerial unconformity (sea-level fall beyond the shelf break as suggested by Dingle et al., 1983) along the shelf. Therefore, we propose that both rivers delivered sediment to the Orange Basin in the Oligocene, but the NW direction may be on account of the strong northerly flowing ocean currents. Ocean currents may have both deflected the rivers northward and triggered the downslope flow of sediment from the shelf break/ uppermost slope.

Lines 492-494: check this sentence. Erosional undercutting is singular, and the sentence reads a little awkwardly.

Reply: Rephrased.

In the section about Miocene Benguela Upwelling, I suggest starting by saying that you are interpreting an analogous situation to the present, and then expand upon this rather than explaining it and then getting to what you are saying.

Reply: This section has been fixed and updated.

The paragraph of line 515 seems redundant to me, as this is not an analogue to what you are describing for the Orange Basin continental slope deposits.

Reply: This has been removed.

Line 525: reference for the overpressured shales?

Reply: Fixed accordingly.

In section 5.4, it is not very clear whether you are suggesting that this deformation is ongoing, or that it took place during deposition and now may get reactivation along its planes of weakness. I think the latter, but please make this clearer?

Reply: We restructured this section and therefore hope things are clearer in the text.

Table 3: Perhaps add which of these units have been sampled – e.g., you have space in the rows below the ages. Alternatively, if there is well / borehole data, add a column for that or state in the caption that all units have been sampled geologically.

Reply: In the deepwater Orange Basin, unfortunately none. No drilling has been conducted there as yet. The table is largely sourced from another study in the Orange Basin (Baby et al., 2018) where well data was available.

Figure 2: please add an inset box of where figure 2 is onto figure 1.

Reply: This has been fixed and updated.

Figure 3: I think this should come ahead of table 3 in the text.

Reply: Table 3 was removed.

Figure 4: as in the methods text, I recommend indicating which of these techniques were done by you, and which were already applied to the data.

Reply: Figure 4 was removed as per Reviewer 1' s suggestion. The interpretation workflow which was done in this study is described in Section 3.2.

Figure 5: Make the text and black lines in panel B bolder and I suggest that the same positions of profiles should also be shown on A. Please also add a small inset of where this is?

Reply: We added the co- ordinates to get the relative position.

Figure 11: Can you link this canyon to a specific river? If so, please name it.

Reply: This is difficult to determine since we would expect a more easterly direction of flow if it were the Orange River when comparing its relative position to the coastline and the trend of the rivers. From the Olifants River's similar NW trend it appears that the canyon was

predominantly fed by it. It is important to note however that the study area lies very far from the coastline and it is unlikely that the canyon extends all the way to the mouth of the rivers as a channel formed as a subaerial unconformity (sea-level fall beyond the shelf break as suggested by Dingle et al., 1983) along the shelf. Therefore, we propose that both rivers delivered sediment to the Orange Basin in the Oligocene, but the NW direction may be on account of the strong northerly flowing ocean currents. Ocean currents may have both deflected the rivers northward and triggered the downslope flow of sediment from the shelf break/ uppermost slope.

Figure 13 is excellent. Please just add a modern coastline position for orientation?

Reply: Added and the entire model updated.

I enjoyed the opportunity to review this manuscript and certainly hope to see the paper published, following revision.

Reply: Thank you very much. We hope so too!

Kind regards.